# DNA methylation-environment interactions in the human genome

Rachel A Johnston[1,2,3]*, Katherine A Aracena[4], Luis B Barreiro[4,5,6], Amanda J Lea[7,8], Jenny Tung[1,8,9,10,11]*

[1]Department of Evolutionary Anthropology, Duke University, Durham, United States; [2]Zoo New England, Boston, United States; [3]Broad Institute of MIT and Harvard, Cambridge, United States; [4]Department of Human Genetics, University of Chicago, Chicago, United States; [5]Section of Genetic Medicine, Department of Medicine, University of Chicago, Chicago, United States; [6]Committee on Immunology, University of Chicago, Chicago, United States; [7]Department of Biological Sciences, Vanderbilt University, Nashville, United States; [8]Canadian Institute for Advanced Research, Toronto, Canada; [9]Duke Population Research Institute, Duke University, Durham, United States; [10]Department of Biology, Duke University, Durham, United States; [11]Department of Primate Behavior and Evolution, Max Planck Institute for Evolutionary Anthropology, Leipzig, Germany

**Abstract** Previously, we showed that a massively parallel reporter assay, mSTARR-seq, could be used to simultaneously test for both enhancer-like activity and DNA methylation-dependent enhancer activity for millions of loci in a single experiment (Lea et al., 2018). Here, we apply mSTARR-seq to query nearly the entire human genome, including almost all CpG sites profiled either on the commonly used Illumina Infinium MethylationEPIC array or via reduced representation bisulfite sequencing. We show that fragments containing these sites are enriched for regulatory capacity, and that methylation-dependent regulatory activity is in turn sensitive to the cellular environment. In particular, regulatory responses to interferon alpha (IFNA) stimulation are strongly attenuated by methyl marks, indicating widespread DNA methylation-environment interactions. In agreement, methylation-dependent responses to IFNA identified via mSTARR-seq predict methylation-dependent transcriptional responses to challenge with influenza virus in human macrophages. Our observations support the idea that pre-existing DNA methylation patterns can influence the response to subsequent environmental exposures—one of the tenets of biological embedding. However, we also find that, on average, sites previously associated with early life adversity are not more likely to functionally influence gene regulation than expected by chance.

## eLife assessment

This **important** paper uses a genome-wide, massively parallel reporter assay to determine how CpG methylation affects regulatory sequences that control the expression of human genes. The authors provide **compelling** evidence that methylation not only influences baseline activity of regulatory sequences but also the magnitude of acute responses to environmental stimuli. The findings are of broad interest, and the extensive data set will likely become a key resource for the community.

## Introduction

DNA methylation is sensitive to the environment, can remain stable across cell divisions, and in some contexts, can alter gene regulation. Consequently, it has received a high level of attention as

**\*For correspondence:**
racheljohnston7@gmail.com (RAJ);
jtung@eva.mpg.de (JT)

a potential pathway linking environmental exposures to downstream trait variation, especially when these exposures are separated temporally, as proposed by the 'biological embedding' hypothesis (*Hertzman, 1999*; *Hertzman, 2012*). In a canonical example, gestational manipulation of methyl donors in mice is causally implicated in changes in DNA methylation and expression levels of the *agouti* gene, which in turn affects susceptibility to obesity and insulin resistance later in life (*Dolinoy et al., 2006*). However, the mechanism underlying the *agouti* case—'metastable' methylation of an inserted intracisternal A particle [IAP] class retrovirus—is now thought to be unusual (*Kazachenka et al., 2018*). Thus, it remains unclear whether the many reports of environment-DNA methylation associations are likely to represent links along a causal pathway to phenotype, or are better viewed as passive biomarkers of exposure.

Experimental tests of the causal impact of DNA methylation on gene regulation indicate that both scenarios are possible. In vitro studies reveal that transcription factor binding is frequently (but not ubiquitously) sensitive to DNA methylation (*O'Malley et al., 2016*; *Yin et al., 2017*). More recently, epigenomic editing of DNA methylation marks has been shown to alter the activity of transcription factors important in disease and development in vivo (*Greenberg and Bourc'his, 2019*; *Yim et al., 2020*), as well as the formation of CTCF-mediated gene loops (*Lei et al., 2017*). However, other epigenetic editing studies show that, even within the same regulatory element, changes in DNA methylation have functional consequences for gene expression at only a subset of sites (e.g. *Maeder et al., 2013*). Indeed, in the native genome, enhancer activity appears to be insensitive to DNA methylation levels in the majority of cases (*Kreibich et al., 2023*). Time series analyses in human dendritic cells have also shown that, in response to a pathogen challenge, nearly all changes in gene expression precede changes in DNA methylation (*Pacis et al., 2019*). Such methylation changes may nevertheless be functionally relevant if they influence the speed and magnitude of subsequent challenges (e.g. by marking the accessibility of latent enhancers across cell divisions) (*Kamada et al., 2018*; *Sun and Barreiro, 2020*). This hypothesis may explain, for example, enhanced transcriptional responses to glucocorticoid stimulation in the descendants of hippocampal progenitor cells previously challenged with glucocorticoids (*Provençal et al., 2020*).

Together, these findings suggest that environment-associated DNA methylation marks are mixed sets that include functionally silent sites, sites that are constitutively important for gene regulation, and sites in which DNA methylation levels affect gene regulation only under certain conditions. Disentangling which sites belong to which set is important for interpreting and prioritizing the results of a growing body of studies in the biological, social, and health sciences, especially those that test hypotheses that assume a functional role for DNA methylation, such as biological embedding (*Hertzman, 2012*; *Demetriou et al., 2015*; *Aristizabal et al., 2020*). To facilitate this process, we recently introduced a massively parallel reporter assay, mSTARR-seq, that is capable of testing the functional effects of DNA methylation on regulatory activity in high-throughput (*Lea et al., 2018*). This method allowed us to test for methylation-dependent regulatory activity at millions of CpG sites in the human genome. However, our previous data set did not investigate how DNA methylation marks affect the response to environmental challenge. It also did not explicitly target the CpG sites that are the most commonly assayed in the human genome (i.e. those featured on the Illumina MethylationEPIC array).

Here, we address these gaps by using mSTARR-seq to construct a genome-wide map of DNA methylation-dependent enhancer activity for 27.3 million CpG sites in the human genome, accomplished by assessing 600 bp windows encompassing 93.5% of CpGs in the human genome as a whole. This set includes 99.3% of CpG sites present on the Illumina MethylationEPIC array, the most commonly used platform for profiling DNA methylation in humans, and 99.4% of CpG sites accessible via reduced-representation bisulfite sequencing (RRBS; *Meissner et al., 2005*). To evaluate the importance of DNA methylation in responses to environmental challenge, we performed mSTARR-seq under baseline conditions (i.e. no challenge) and following exposure to two physiologically relevant environmental challenges: dosage with the synthetic glucocorticoid dexamethasone, which plays a central role in metabolic regulation and stress-related homeostasis, and interferon alpha (IFNA), a cytokine that elicits genomic and immunological responses associated with viral infection.

We used these data to pursue three goals. First, we describe overall patterns of regulatory activity and methylation-dependent regulatory activity across profiled sites, including how sites targeted by the EPIC array or by RRBS compare to the genome as a whole. The results of this analysis produce a new resource: a map of DNA methylation-dependent enhancer activity across the human genome.

Second, we test the degree to which methylation-dependent regulatory activity is affected by exposure to steroid hormone or immune defense-related signaling molecules. Our results yield insight into the frequency of DNA methylation-environment interactions genome-wide, and provide support for the hypothesis that DNA methylation potentiates the cellular response to external stressors. Finally, we illustrate the applicability of this resource by testing two predictions of the biological embedding hypothesis: that pre-existing DNA methylation levels can affect the transcriptional response to subsequent environmental challenge (*Provençal et al., 2020*; *Sun and Barreiro, 2020*; *Fanucchi et al., 2021*), and that sites linked to early life adversity (ELA) are likely to be functionally important for shaping gene expression. Our results suggest that mSTARR-seq can identify DNA methylation-environment interactions that also occur in vivo in human populations and can therefore help prioritize environment- and ELA-associated loci for future follow-up.

## Results
### mSTARR-seq captures enhancer activity and methylation-dependent enhancer activity genome-wide

We assessed regulatory activity and methylation-dependent regulatory activity for 4,558,475 600-base pair windows of the genome by pairing hybridization capture of targeted loci in the human genome with the massively parallel reporter assay, mSTARR-seq (*Lea et al., 2018*). In brief, mSTARR-seq performs enzymatic manipulation of DNA methylation across hundreds of thousands to millions of reporter DNA fragments simultaneously. By measuring the ability of fragments to self-transcribe (as in *Arnold et al., 2013*; *Shlyueva et al., 2014*; reviewed in *Gallego Romero and Lea, 2022*), it generates estimates of the enhancer-like regulatory potential for each fragment when that fragment is in an unmethylated versus a methylated state (*Figure 1A, B*). Notably, results from the unmethylated condition are akin to those from a conventional STARR-seq experiment, in that they assess regulatory activity irrespective of CpG content or methylation status. To focus on the CpG sites most likely to be included in DNA methylation studies in humans, we performed custom capture with SeqCap EZ Prime Choice Probes (Roche), targeting all CpG sites on the Illumina Infinium MethylationEPIC array and those likely to be profiled using reduced representation bisulfite sequencing, which enriches for CpG sites near targets of *MspI* restriction enzyme digestion (*Figure 1—figure supplement 1*). Because of substantial attention to potential environmental and early life effects on DNA methylation at the glucocorticoid receptor gene (*NR3C1*, reviewed in *Liu and Nusslock, 2018*), we also targeted the 6.5 Mb in and flanking *NR3C1*. Finally, to assess background expectations for regulatory and methylation-dependent regulatory activity in the human genome, we targeted a control set of 100,000 loci for capture, chosen at random across the human genome after excluding centromeres, gaps, and uncalled bases in the hg38 genome.

We generated an mSTARR-seq library from captured DNA from the GM12878 cell line, followed by transfection into the K562 erythroleukemic cell line and co-purification of plasmid-derived RNA and DNA (6 replicates per condition: methylated versus unmethylated; see Materials and methods; *Supplementary file 1*). Based on our minimal criteria for assessment (600 basepair windows where at least half of the plasmid-derived DNA samples had non-zero reads covering the window; see Materials and methods), 90.2% of the human genome was included in the plasmid DNA library purified at the end of the mSTARR-seq experiment (mean sequencing depth per replicate = 30.4 million reads; mean read coverage per window = 12.723 ± 41.696 s.d.). Within this set, which also included off-target regions, windows were ~15 fold enriched for targeted regions in the genome (Fisher's Exact Test $\log_2$(OR) 95% confidence interval (CI)=3.971 [3.934, 4.009], p<1.0 x $10^{-300}$). Thus, in windows that passed our minimal assessment criteria, we successfully targeted 99.3% of CpG sites on the MethylationEPIC array and 99.4% of sites likely to be included in RRBS libraries. Across target sets, read depth was not predicted by CpG density, indicating no systematic power differences in assessing methylation-dependent activity due to differences in CpG number per window ($R^2$=0.099, p=0.6852; *Supplementary file 2*). Our results are comparable to published fragment diversity levels achieved with this method (*Lea et al., 2018*; *Figure 1—figure supplement 2*). Because demethylation or remethylation of DNA fragments could occur within cells during the experiment, we performed bisulfite sequencing at the end of the experiment. We confirmed that DNA methylation levels were substantially higher in the methylated condition samples than in the unmethylated samples, where methylation levels were

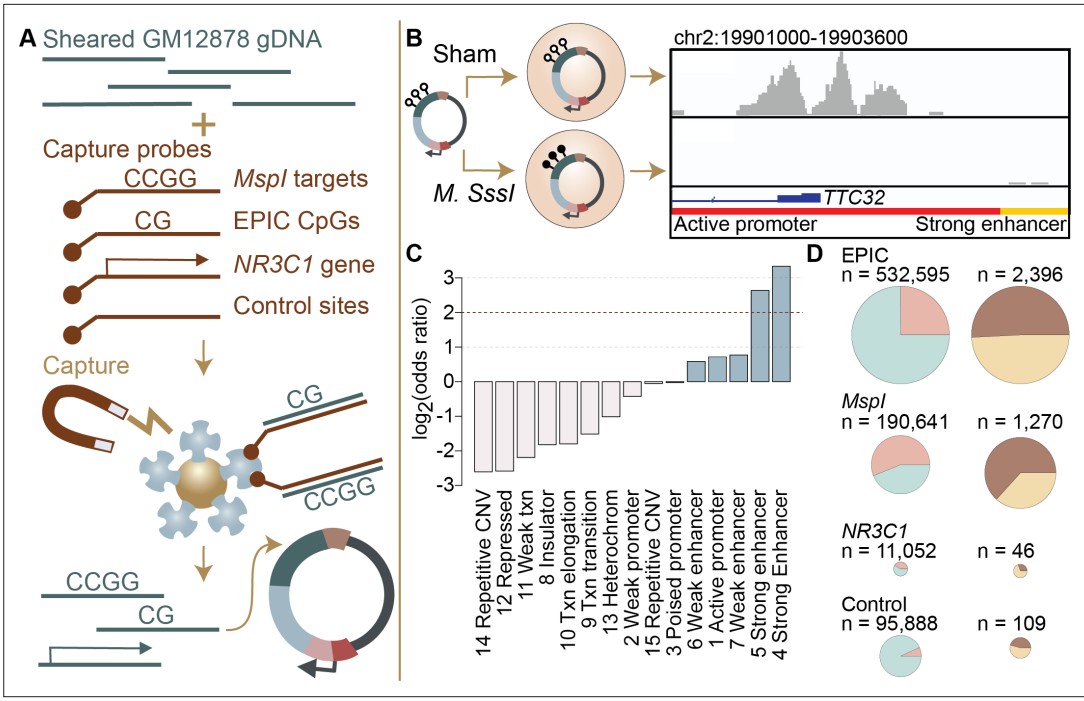

**Figure 1.** Study design and identification of enhancer activity. (**A**) Sheared DNA from the GM12878 cell line was subjected to enrichment via capture with probes targeting loci selected in reduced representation bisulfite sequencing (RRBS) workflows (*MspI* targets), CpG sites on the Infinium EPIC array, the gene *NR3C1* and flanking regions, and 100,000 randomly distributed control regions. Note that a single 600 bp window can contain multiple target types (*Figure 1—figure supplement 1*). Library diversity summaries are shown in *Figure 1—figure supplement 2*. (**B**) Captured loci were cloned into the mSTARR-seq vector, *pmSTARRseq1*, treated with either the CpG methylating enzyme *M. SssI* or a sham treatment, and transfected into K562 cells. Methylation levels post-transfection and rarefaction analyses of sequencing depth of replicate samples are shown in *Figure 1—figure supplements 3 and 4*. Right panel shows an example of DNA methylation-dependent regulatory activity near the first exon of the *TTC32* gene, where the methylation-dependent regulatory element overlaps an active promoter chromatin state (red horizontal bar denotes active promoter as defined by ENCODE: *The ENCODE Project Consortium, 2012*). (**C**) mSTARR-seq regulatory activity in the baseline condition is strongly enriched in ENCODE-defined enhancers and some classes of promoters (indicated in blue), and depleted in repressed, repetitive, and heterochromatin states. See *Supplementary file 4* for full results of this analysis. Regions with mSTARR-seq regulatory activity detected in this experiment also significantly overlap with regions with regulatory activity in other mSTARR-seq and conventional STARR-seq datasets (*Figure 1—figure supplement 5*; see also *Figure 1—figure supplement 6* for estimates of concordance across technical replicates). (**D**) Left column shows, under the baseline condition (i.e. unstimulated cells), the proportion of 600 bp windows that exhibited minimal regulatory activity (at least 3 replicate samples produced non-zero RNA-seq reads in either the methylated condition or the unmethylated condition) in the mSTARR-seq assay (pink) versus those with detectable input DNA but no evidence of regulatory activity (blue), for windows containing sites from each target set. Right column shows the proportion of windows with regulatory capacity (i.e., the subset of the windows represented in pink on the left that produce excess RNA relative to the DNA input at FDR <1%) that are also methylation-dependent (dark brown). Within each column, pie charts are scaled by the total numbers of windows represented. See *Figure 1—figure supplements 7 and 8* for comparisons to regulatory regions in other datasets. *Figure 1—figure supplements 9 and 10* show window-level RNA to DNA ratios. *Figure 1—figure supplement 11* shows the relationship between CpG density and methylation-dependent regulatory activity.

The online version of this article includes the following figure supplement(s) for figure 1:

**Figure supplement 1.** Overlap of target genomic regions with each other.

**Figure supplement 2.** mSTARR-seq library diversity.

**Figure supplement 3.** Methylation levels of mSTARR-seq DNA, pre- and post-transfection.

**Figure supplement 4.** Rarefaction curve showing total number of windows formally tested for regulatory activity, as a function of number of reads sequenced per DNA or RNA replicate.

*Figure 1 continued on next page*

*Figure 1 continued*

**Figure supplement 5.** Overlap of regulatory activity across datasets.

**Figure supplement 6.** Correlations between RNA and DNA replicates.

**Figure supplement 7.** Library diversity and regions of regulatory activity in HepG2 cells.

**Figure supplement 8.** Methylation-dependent regulatory activity across datasets.

**Figure supplement 9.** RNA to DNA ratios in the methylated and unmethylated replicates in the baseline dataset.

**Figure supplement 10.** Histograms of RNA to DNA ratios in the baseline dataset.

**Figure supplement 11.** Relationship between CpG density and methylation-dependent regulatory activity.

---

near zero (mean methylated = 0.885, mean unmethylated = 0.066, *Figure 1—figure supplement 3*; unpaired t-test: t=–14.66, df = 15,124, p=2.39 x $10^{-10}$). We note that the observed small deviations from 0% and 100% methylation would reduce our power to detect methylation-dependent activity, but should not incur false positive results.

To test for regulatory potential, we focused on those windows where at least three replicate samples produced non-zero RNA-seq reads in either the methylated condition or in the unmethylated condition (n=216,091 non-overlapping 600 bp windows, 4.3% of the human genome; mean DNA-seq read coverage per window = 93.907 ± 10.091 s.d.; mean RNA-seq read coverage per window = 165.093 ± 1008.816). These inclusion criteria focused our analysis on the subset of windows where we were able to detect minimal evidence of transcription from the plasmid. The proportion of the human genome showing evidence of regulatory activity in STARR-seq family assays is expected to be small based on previous work. For example, Johnson and colleagues estimated that 0.165% of the human genome had regulatory potential in unstimulated A549 cells in a genome-wide STARR-seq assay, based on the proportion of hg38 basepairs that fell in regions with regulatory activity in their assay (*Johnson et al., 2018*). Among the regions analyzed here, 3721 windows (1.7%) showed evidence for regulatory activity, where the amount of RNA generated from a window significantly exceeded the amount of DNA input sequenced for the same window (FDR <1%; *Supplementary files 3*; 0.082% of all windows where DNA reads were obtained in at least half of our replicate samples). Sequencing depth does not appear to constrain the ability to detect regulatory regions, as rarefaction analyses suggest that increasing sequencing depth of RNA or DNA libraries would add only a small number of additional regions with any evidence of transcriptional activity (*Figure 1—figure supplement 4*). Further, among analyzed windows, DNA coverage per fragment explains minimal variance in mSTARR-seq enhancer activity ($R^2$=0.029, p<1.0 x $10^{-300}$).

Regions with regulatory activity were enriched for enhancers and active promoters annotated by the ENCODE Consortium for the K562 cell line log$_2$(OR)=0.584–3.337; all p<1 x $10^{-3}$; *Figure 1C*; *Supplementary file 4* (*The ENCODE Project Consortium, 2012*; *Ernst and Kellis, 2012*); note that previous research has found that many promoters can show enhancer-like activity in STARR-seq-type assays, although some promoters will be missed (e.g. *Klein et al., 2020*). In support of assay reproducibility, regulatory regions also were highly consistent with previous mSTARR-seq data generated in K562 cells using a previously published, independently generated DNA library that we reanalyzed using the current pipeline (overlap among windows with FDR <1% in each data set analyzed independently: log$_2$(OR) [95% CI]=6.212 [6.086, 6.440], p<1.0 x $10^{-300}$; *Figure 1—figure supplements 5–6*; *Supplementary files 5-6*; *Lea et al., 2018*).

Regions with mSTARR-seq-annotated regulatory activity also exhibited a high degree of consistency between cell types. In a comparison against newly generated mSTARR-seq data from the HepG2 liver cell line, using the same mSTARR-seq library as in the previously published K562 experiments (*Lea et al., 2018*), regulatory regions detected in this study were enriched among regulatory regions detected in HepG2 cells (log$_2$(OR) [95% CI]=3.534 [3.381, 3.684], p=5.21 x $10^{-307}$; *Figure 1—figure supplements 5 and 7*; *Supplementary file 7*). They also overlap with regulatory regions identified in a conventional STARR-seq experiment in A549 lung epithelial cells (log$_2$(OR) [95% CI]=2.451 [2.442, 2.461], p<1.0 x $10^{-300}$) (*Johnson et al., 2018*). Although we prioritized a blood-derived cell line because of the frequency of environmental epigenetic studies done in blood, these results suggest that our findings can be generalized, to some extent, to other cell and tissue types.

Among the 3721 windows with regulatory activity in either the methylated or unmethylated condition, 1768 windows (47.5% of regulatory windows; FDR <1%) were differentially active depending on

condition, pointing to DNA methylation-dependent regulatory activity. This result is also concordant when we apply the current pipeline to previously published data from K562s and our new HepG2 data, which exhibit correlated effects of DNA methylation on regulatory activity in the windows examined in both cases (K562: $R^2$=0.286 for 1250 regulatory windows with FDR <1% in both data sets, p=3.19 x $10^{-93}$; HepG2: $R^2$=0.277 for 511 regulatory windows with FDR <1% in both data sets, p=8.87 x $10^{-38}$ *Figure 1—figure supplement 8*). Among methylation-dependent regulatory regions, the majority of regions (1744 windows, 98.6%) were more active in the unmethylated condition than the methylated condition (*Figure 1—figure supplements 9 and 10*). Consistent with previous findings (*Lea et al., 2018*), regulatory regions with more CpGs are more likely to be repressed by DNA methylation (*Figure 1—figure supplement 11*; Spearman's rho = 0.370, p=9.865 x $10^{-121}$; n=3,721 regions with mSTARR-seq regulatory activity). Regulatory windows showing higher activity in the methylated condition (24 windows) were enriched for binding motifs of the transcription factors p53, which has been previously reported to have increased binding affinity to methylated DNA relative to unmethylated DNA (*Kribelbauer et al., 2017*) ($\log_2$(OR) [95% CI]=2.352 [0.714, 3.767], p=2.91 x $10^{-3}$; *Supplementary file 8*), and Tfcp2I1, which has been found to recruit Tet2 to mediate enhancer demethylation (*Sardina et al., 2018*) ($\log_2$(OR) [95% CI]=2.615 [0.580, 4.229], p=6.84 x $10^{-3}$).

## Commonly studied CpG sites are enriched for methylation-dependent regulatory activity

We compared methylation-dependent regulatory activity between CpGs targeted on the EPIC array, CpGs typically profiled in RRBS libraries (i.e. near *MspI* cut sites), the 6.5 Mb in or flanking the glucocorticoid receptor gene *NR3C1*, and the control set of genome-representative loci. As expected, windows that contained EPIC CpG sites, sites associated with RRBS, or that were located in or near *NR3C1* were significantly more likely to show at least some degree of transcriptional activity (i.e. show plasmid-derived RNA reads in at least half the sham or methylated replicates) than the genomic background (i.e. 98,967 randomly distributed loci: see Materials and Methods), and thus were more likely to be included in our analysis set (n=216,091 windows) for regulatory activity (EPIC: $\log_2$(OR) [95% CI]=2.189 [2.152, 2.227], p<1.0 x $10^{-300}$; RRBS: $\log_2$(OR) [95% CI]=4.126 [4.086, 4.163], p<1.0 x $10^{-300}$; *NR3C1*: $\log_2$(OR) [95% CI]=3.259 [3.193, 3.325], p<1.0 x $10^{-300}$; *Figure 1D*). However, conditional on minimal transcriptional activity, windows containing EPIC array sites were not more likely to exhibit *significant* enhancer-like regulatory activity (FDR <1%; n=3721 windows) than background windows of the genome, and RRBS- and *NR3C1*-associated sites were in fact slightly less likely to do so (EPIC: $\log_2$(OR) [95% CI]=0.112 [-0.168, 0.405], p=0.47; RRBS: $\log_2$(OR) [95% CI]=−0.493 [-0.779,−0.195], p = 1.11 x $10^{-3}$; *NR3C1*: $\log_2$(OR) [95% CI]=−0.730 [-1.263,−0.217], p = 3.89 x $10^{-3}$). This result is likely explained by our inclusion criteria, as only regions with evidence for minimal RNA transcription were retained prior to formal analysis.

In contrast, among regions with detectable regulatory activity, 63.3% of those containing RRBS-associated CpG sites exhibited methylation-dependent regulatory activity, compared to 45.9% of control background loci (RRBS against control: $\log_2$(OR) [95% CI]=1.025 [0.430, 1.624], p=4.39 x $10^{-4}$; *Figure 1D*). Note that of 109 regulatory windows in the control set, 108 contain at least 1 CpG, so this difference is not because the control set is CpG-free (mean number CpGs per fragment in control set = 13.291 ± 11.179 s.d.; EPIC = 19.641 ± 14.471; RRBS = 20.870 ± 13.883; *NR3C1*=7.405 ± 9.409). Neither windows with EPIC CpGs nor windows in or near *NR3C1* were enriched for methylation-dependent regulatory activity relative to the genomic background (EPIC against control: $\log_2$(OR) [95% CI]=0.284 [-0.297, 0.871], p=0.33; *NR3C1* versus control: $\log_2$(OR) [95% CI]=−0.948 [-2.129, 0.178], p=0.11). Consequently, differential methylation identified through RRBS is more likely to be capable of driving differences in gene regulation, as detectable by mSTARR-seq, than differential methylation elsewhere in the genome. We caution, however, that even among RRBS sites, 98.8% do not occur in regions of the genome with detectable regulatory activity in either the methylated or unmethylated conditions, at least in the K562 cell type, and 35.0% of those that fall in putative regulatory elements exhibit no evidence for methylation-dependent activity.

## Environmental perturbation reveals cryptic regulatory elements and cryptic effects of DNA methylation

Enhancer activity can be cell type- or environment-dependent (e.g. *Ostuni et al., 2013*; *Johnson et al., 2018*; *Chaudhri et al., 2020*). DNA methylation-dependent enhancer activity may show similar context-dependence, thus potentially accounting for the large number of sites that fall in functionally silent regions described above. However, this possibility has not been systematically tested. To do so, we next compared regulatory activity between the baseline unchallenged condition and cells challenged with interferon alpha (IFNA) or dexamethasone (dex) (*Figure 2A*; *Figure 2—figure supplements 1–2*; *Supplementary file 9*). Regions that were identified to have regulatory potential in the baseline condition were highly likely to retain regulatory potential after cells were challenged with IFNA or dex (*Figure 2—figure supplement 3*; IFNA $\log_2(OR)$ [95% CI]=8.639 [8.499, 8.812], $p<1.0 \times 10^{-300}$; dex $\log_2(OR)$ [95% CI]=9.640 [9.483, 9.776], $p<1.0 \times 10^{-323}$; *Supplementary files 10-11*). However, environmental challenges also revealed thousands of putative regulatory regions that were undetectable at baseline but active post-stimulation (*Figure 2B*; 1614 IFNA-specific; 1131 dex-specific). Of 4632 IFNA regulatory regions (<1% FDR), 44.1% are not detectable at baseline at a 1% FDR threshold in the baseline condition, and even with a relaxed baseline FDR of 10%, 25.1% remain undetectable. Of 4217 dex regulatory regions (<1% FDR), 40.2% are not detectable at baseline (1% FDR), and 31.5% remain undetectable at a baseline FDR of 10%.

Regulatory windows specific to the IFNA treatment were enriched for 33 transcription factor binding motifs (TFBMs) (*Supplementary file 12*), with strong enrichments detected for TFBMs involved in innate immune defense in general, and interferon signaling specifically (*Figure 2C*). For example, the most enriched motif was the canonical DNA target of interferon signaling, known as IFN-stimulated response elements (ISRE; $\log_2(OR)$ [95% CI]=3.158 [2.953, 3.358], Bonferroni corrected $p=7.31 \times 10^{-133}$), followed by binding motifs for several IFN-regulatory factors (IRF1, IRF2, IRF3, IRF4, IRF8; all OR >1.5 and Bonferroni corrected $p<1 \times 10^{-15}$; *Chen et al., 2017*). Regulatory windows specific to the dex-stimulated condition were significantly enriched for 28 transcription factor binding motifs, including the glucocorticoid response element IR3 and binding motifs of several transcription factors known to interact with or be modulated by the glucocorticoid receptor (AP-1, CEBP:CEBP, CEBP:AP1, JunB, Jun-AP1, GATA1, STAT3, and STAT5; all OR >1.3, Bonferroni corrected $p<0.01$; *Supplementary file 13*; *Cain and Cidlowski, 2017*). Results were qualitatively similar if we used a more stringent definition of IFNA-specific and dex-specific regulatory activity (e.g. 'IFNA-specific' defined as FDR <1% in IFNA condition and FDR >10% in the other two conditions; *Supplementary files 14-15*).

To evaluate the relevance of these regions to in vivo gene regulation, we also generated matched mRNA-seq data for the endogenous K562 genome from the same experiments. These data showed that expressed genes located closest to mSTARR-seq-annotated, IFNA-specific enhancers were more strongly upregulated after IFNA stimulation than the set of expressed genes located closest to shared mSTARR-seq-annotated enhancers (i.e. those identified in both the IFNA and at least one other condition, considering genes within 100 kb maximum distance from each enhancer element; unpaired t-test; t=3.268, df = 601.88, $p=1.15 \times 10^{-3}$). We found similar results when using inferred enhancer-gene linkages from the EnhancerAtlas 2.0 (*Gao et al., 2016*) and restricting the set of IFNA-specific enhancers to those with external experimental support for ISRE binding (n=1005 windows; ChIP-Seq data from *The ENCODE Project Consortium, 2012*, relative to all other genes in the gene expression dataset; unpaired t-test; t=3.579, df = 118.36, $p=5.01 \times 10^{-4}$; *Figure 2D*; *Supplementary file 16*). Despite a weaker overall gene expression response to dex, genes closest to dex-specific mSTARR-seq enhancers were also more strongly upregulated after dex stimulation than genes near shared enhancers (unpaired t-test; t=3.477, df = 479.6, $p=5.53 \times 10^{-4}$).

For the IFNA challenge, the DNA methylation state of each window appears to play an important role in shaping condition-specific responses to stimulation in the mSTARR-seq data set. Nearly twice as many regulatory windows (81.4%; 1314 of 1614) exhibited methylation-dependent regulatory activity in the IFNA-specific condition than in the baseline or dex-specific condition (47.5% and 48.4% respectively; two-sided binomial test for IFNA compared to baseline: $p=3.58 \times 10^{-312}$). Further, regulatory regions that harbor TFBMs for TFs central to the interferon response (ISRE, IRF1, IRF2, IRF3, IRF4, IRF8; N=663 windows) strongly responded to IFNA challenge if in an unmethylated state, but mounted systematically attenuated responses if in a methylated state. As a result, 562 of these 663 windows (84.7%) exhibited significant methylation-dependent regulatory activity and 561 of them

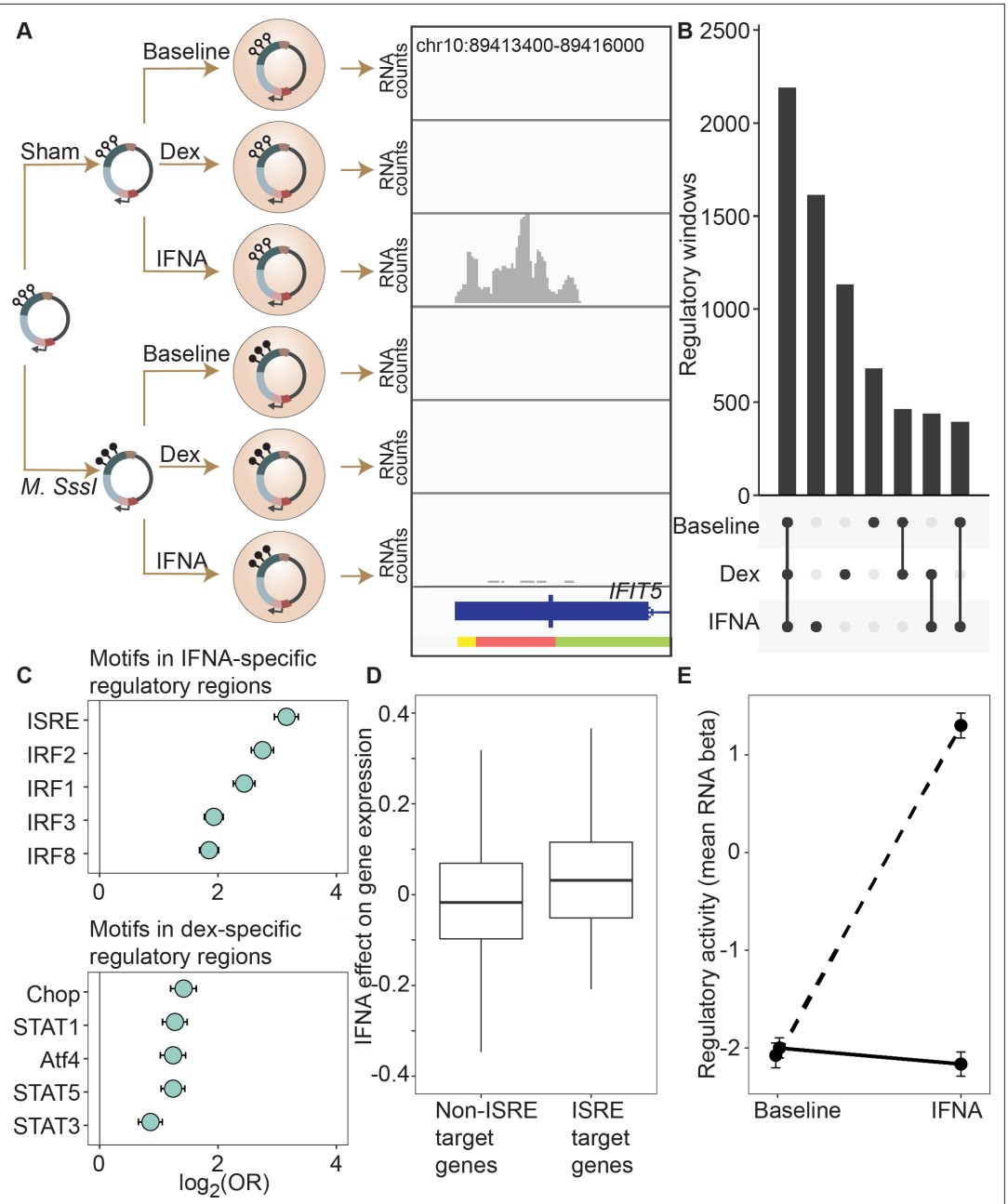

**Figure 2.** DNA methylation-environment interactions reveal methylation-dependent responses to IFNA and dexamethasone challenge. (**A**) Full mSTARR-seq design across DNA methylation and challenge conditions (see *Figure 2—figure supplements 1 and 2* for filtering and overlap of the datasets, and *Supplementary files 3, 10 and 11* for effect sizes). An example of a DNA methylation-environment interaction is shown overlapping the interferon-induced gene *IFIT5* and an ENCODE-annotated weak promoter (pink denotes weak promoter, yellow denotes heterochromatin, and green denotes weak transcription; the endogenous *IFIT5* gene expression response to IFNA in our experiment is shown in *Figure 2—figure supplement 4*). Three consecutive 600 bp windows have interaction FDR $<1 \times 10^{-4}$ in this region. Panels depict non-normalized, raw read pileups for mSTARR-seq RNA replicates, with all y-axis maximums set to 14,000. No methylation-dependent activity is detectable in the baseline condition because this enhancer element is inactive. Upon IFNA stimulation, only unmethylated enhancer elements are capable of responding. (**B**) Upset plot showing shared and unique mSTARR-seq identified enhancer elements across conditions. While many elements are shared, 3426 are unique to a single condition (FET log-odds results for magnitude of overlap are shown in *Figure 2—figure supplement 3*). (**C**) Top five most enriched transcription factor binding motifs in IFNA- and dex-specific mSTARR-seq enhancers, compared to all windows

*Figure 2 continued on next page*

*Figure 2 continued*

tested. Whiskers show the 95% CI. See **Supplementary files 12 and 13** for all enrichment results. (**D**) Genes targeted by ISRE enhancers (ISRE enhancers identified from ENCODE ChIP-seq data; gene targets identified from enhancer-gene linkages from EnhancerAtlas 2.0: **Gao et al., 2016**) that are also identified as IFNA condition-specific mSTARR-seq enhancers (n=119) show stronger K562 endogenous gene expression responses to IFNA stimulation than non-ISRE targets (n=10557; unpaired t-test: t=3.58, df = 118.36, p=5.01 x 10$^{-4}$; **Supplementary file 16**). Each box represents the interquartile range, with the median value depicted as a horizontal bar. Whiskers extend to the most extreme values within 1.5 x of the interquartile range. (**E**) mSTARR-seq regulatory activity for windows containing ISRE targets (n=1,005 windows) interacts strongly with exposure to IFNA. These windows are capable of mounting a strong response to IFNA stimulation when unmethylated (dashed line; paired t-test: t=23.02, df = 1004, p=1.78 x 10$^{-94}$) but not when methylated (solid line; paired t-test: t=–1.74, df = 1004, p=0.082). Dots show the mean beta corresponding to enrichment of RNA reads versus DNA reads across windows; whiskers show the standard error. Because y-axis values correspond to model estimates, they can be positive (i.e. more mSTARR-seq RNA reads than input DNA reads) or negative values (i.e. fewer mSTARR-Seq RNA reads than mSTARR-Seq input DNA reads, indicating no regulatory activity).

The online version of this article includes the following figure supplement(s) for figure 2:

**Figure supplement 1.** Filtering results across datasets.

**Figure supplement 2.** Overlap of tested genomic windows across datasets.

**Figure supplement 3.** Overlap of regulatory activity and effects of methylation across environmental conditions.

**Figure supplement 4.** *IFIT5* endogenous gene expression is responsive to IFNA stimulation.

---

(99.8%) were more active in the IFN-challenged state when unmethylated. This pattern is recapitulated when focusing on analyzed windows with external experimental support for ISRE binding (n=1005 windows; ChIP-Seq data from **The ENCODE Project Consortium, 2012**). These windows show no evidence for methylation-dependent regulatory activity in the baseline condition (paired t-test; t=–0.792, df = 1004, p=0.43), primarily because they show no regulatory activity at all without IFNA stimulation. After IFNA stimulation, though, they exhibit strong methylation-dependence. Specifically, only unmethylated windows are capable of a response (paired t-test; t=31.748, df = 1004, p=1.02 x 10$^{-153}$; **Figure 2E**).

## Methylation levels at mSTARR-seq IFNA-specific enhancers predict the transcriptional response to influenza virus in human macrophages

Our findings show that, in the context of mSTARR-seq, pre-existing DNA methylation state can capacitate or constrain the regulatory response to IFNA stimulation. This result suggests that DNA methylation-environment interactions may be an important determinant of gene expression levels in vivo. To test this possibility, we drew on matched whole genome bisulfite sequencing (WGBS) and RNA-seq data collected from human monocyte-derived macrophages (n=35 donors), with and without infection with the influenza A virus (IAV), commonly known as flu (**Figure 3A**; **Aracena et al., 2024**). We asked whether variation in the gene expression response to flu is predicted by DNA methylation levels in the baseline (non-infected) condition, especially at loci where the response to IFNA is affected by DNA methylation in mSTARR-seq. Importantly, flu and IFNA challenges induce similar innate immune responses (**Killip et al., 2015**).

Within each individual in the macrophage data set, mean DNA methylation levels at baseline significantly predict the mean gene expression response to flu across the full set of mSTARR-seq enhancer windows detected in the IFNA condition: higher methylation at baseline predicts an attenuated gene expression response, on average (n=35 individuals at 2769 enhancer windows: mean Pearson's *r*=–0.105 ± 0.006 s.d., all Bonferroni-corrected p<3 x 10$^{-5}$; **Supplementary file 17**). This effect is largely driven by the subset of mSTARR-seq IFNA-specific enhancers. Specifically, the correlation between baseline DNA methylation levels and the gene expression response to flu is 3.44-fold stronger in IFNA-specific enhancers than for enhancers identified in both the IFNA condition and at least one other condition (**Figure 3B**; IFNA-specific enhancers: n=1033, mean Pearson's *r*=–0.170 ± 0.009 s.d., all Bonferroni-corrected p<2 x 10$^{-5}$; shared enhancers: n=1736, mean Pearson's *r*=–0.049 ± 0.01 s.d., all Bonferroni-corrected p>0.1). These results appear to be driven by strong methylation-dependence in flu-infected cells, as the average within-individual correlation between DNA methylation and gene

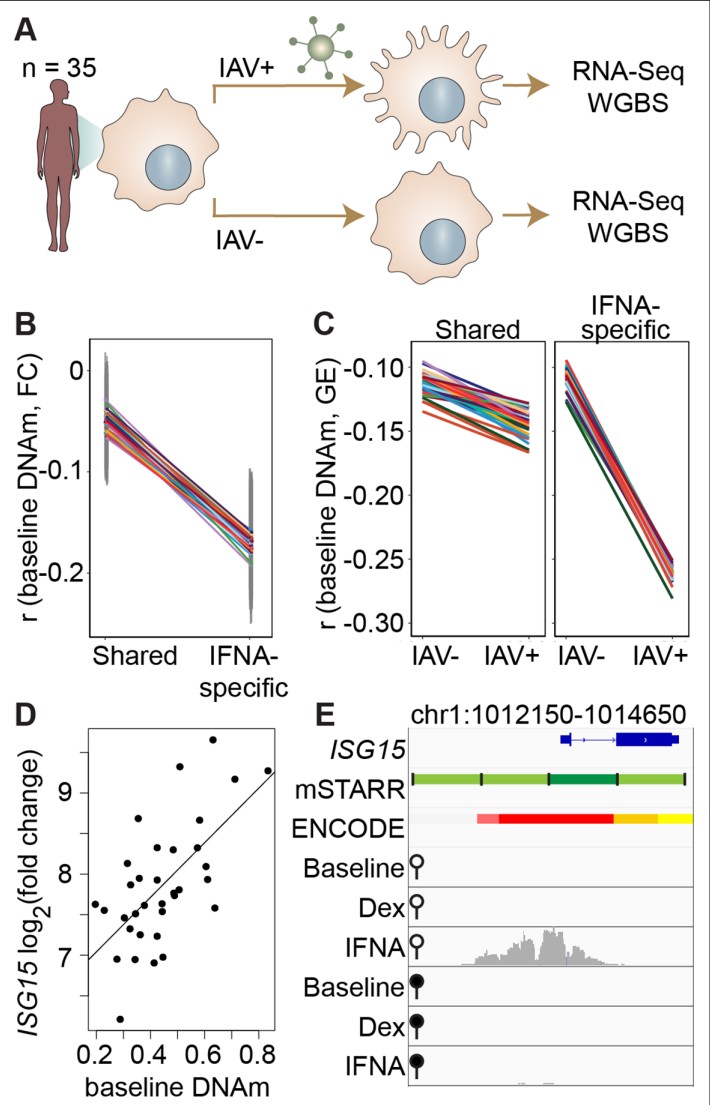

**Figure 3.** DNA methylation in mSTARR-seq enhancers predicts in vivo gene expression in macrophages. (**A**) Study design of the in vivo experiment, in which matched macrophage samples from 35 individuals were either left non-infected or infected with influenza A virus (IAV) for 24 hours and processed for RNA-seq and whole genome bisulfite sequencing (WGBS; **Aracena et al., 2024**). (**B**) Within individuals, DNA methylation (DNAm) levels at mSTARR-seq enhancers in non-infected cells are negatively correlated with the nearest genes' transcriptional responses to IAV, but only in mSTARR-seq enhancers that were specific to the IFNA condition (IFNA-specific enhancers: n=1033, mean Pearson's $r$=–0.170 ± 0.009 s.d., all Bonferroni-corrected $p<2 \times 10^{-5}$; shared enhancers: n=1736, mean Pearson's $r$=–0.049 ± 0.01 s.d., all Bonferroni-corrected $p>0.1$). Each colored line represents an individual, and vertical gray lines represent 95% confidence intervals (see **Supplementary file 17** for full results). (**C**) The average within-individual correlation (**r**) between DNA methylation and gene expression (GE) is 2.44 times as large after infection ($r$=–0.261 ± 0.006) than at baseline ($r$=–0.106 ± 0.008) in IFNA-specific mSTARR-seq enhancers (right panel), but much less affected by infection at mSTARR-seq enhancers that are shared across conditions (left panel). Within each panel, each colored line represents an individual for the same set of enhancers (see **Supplementary file 18** for full results), without and with IAV infection. (**D**) Across individuals, the *ISG15* transcriptional response to IAV is significantly correlated with average DNAm at the mSTARR-seq enhancer chr1:1013400–1014000 in non-infected cells ($R^2$=0.381, $p$=6.05 x $10^{-5}$, q=0.084). Each dot represents an individual (see **Supplementary file 19** for full results and **Figure 3—figure supplement 1** for condition-specific results). (**E**) The mSTARR-seq enhancer predictive of *ISG15* response to IAV (dark green bar) is located in the active promoter of *ISG15* (as defined by ENCODE **The ENCODE Project Consortium, 2012**; red denotes active promoter, pink denotes weak promoter, orange denotes strong enhancer, yellow denotes weak enhancer). Three adjacent, methylation-dependent, IFNA-specific mSTARR-seq enhancers were identified (light green), but do not

*Figure 3 continued on next page*

*Figure 3 continued*

significantly predict *ISG15* response to IAV (q>10%). The bottom 6 tracks depict non-normalized, raw read pileups for mSTARR-seq RNA replicates in either the unmethylated (open circle) or methylated (filled circle) condition, with all y-axis maximums set to 20,000.

The online version of this article includes the following figure supplement(s) for figure 3:

**Figure supplement 1.** Across individuals, methylation in the mSTARR-seq annotated enhancer chr1:1013400–1014000 predicts the *ISG15* gene expression response to flu.

expression is 2.44 times as large after infection ($r$=–0.261 ± 0.006) than at baseline ($r$=–0.106 ± 0.008) in IFNA-specific mSTARR-seq enhancers (*Figure 3C*; *Supplementary file 18*).

The limited sample size of the macrophage data set makes it better suited for analyses of the overall relationship between DNA methylation and gene expression within an individual, rather than locus-specific analyses of interindividual variation (especially as locus-specific variance in DNA methylation across individuals is low: mean = 0.004, standard deviation = 0.008). Nevertheless, across 1382 testable loci (600 bp windows containing at least 1 CpG with interindividual variance >0.01), we identified one IFNA-specific, methylation-dependent mSTARR-seq enhancer where endogenous variation in DNA methylation levels across individuals clearly predicts the response of the nearest gene's transcriptional response to flu (*Figure 3D–E*; p=6.05 x 10$^{-5}$, q-value=0.0837; *Supplementary file 19*). This mSTARR-seq enhancer (chr1:1013400–1014000) overlaps the promoter of interferon-stimulated gene 15 (*ISG15*; *Figure 3E*), and its average methylation explains 38% of the variance in the *ISG15* transcriptional response to flu across individuals. The magnitude of *ISG15* transcriptional response to flu appears primarily driven by the enhancer's effect on gene expression at baseline. Lower enhancer methylation at baseline is associated with higher *ISG15* expression at baseline, ultimately resulting in a shallower fold change response to flu (*Figure 3—figure supplement 1*). This example illustrates the value of integrating observational data on DNA methylation and gene expression with mSTARR-seq. The population data support the in vivo relevance of the mSTARR-seq data, while the mSTARR-seq data suggest that baseline variation in *ISG15* methylation in vivo is causally meaningful to the response to influenza. Notably, *ISG15* plays critical roles in regulating the type I interferon response and modulating host immunity to both viral and bacterial infections (reviewed in *Perng and Lenschow, 2018*).

## Most CpG sites associated with early life adversity do not show regulatory activity in K562s

Finally, because changes in DNA methylation are of particular interest to research on the biological embedding of early life experience (*Hertzman and Boyce, 2010*), we tested whether DNA methylation differences associated with early life adversity (ELA) translate to functional effects on gene regulation in the mSTARR-seq data. We first performed a literature search to compile CpG sites that have previously been associated with ELA in humans using the Illumina EPIC array or one of its precursors (Infinium Human Methylation 450K and 27K BeadChips). Our search resulted in a total of 27 studies (*Supplementary file 20*), which together identified 8,525 unique ELA-associated sites.

For 26 of 27 studies, ELA-associated CpG sites were not more likely to occur within putative regulatory windows (detected in either the methylated condition, unmethylated condition, or both) than background chance (*Figure 4*). This pattern was qualitatively consistent regardless of whether we considered baseline, IFNA-, or dex-challenged samples (*Supplementary file 20*). The only exception was for a set of differentially methylated regions in the children of mothers exposed to objective hardship (e.g. living in a shelter, loss of electricity) who were pregnant during or within 3 months of the 1998 Quebec Ice Storm (*Cao-Lei et al., 2014*). In this study, ELA-associated sites were more likely to fall in windows with regulatory potential in our sample ($\log_2$(OR) [95% CI]=1.343 [0.715, 1.909], p=3.39 x 10$^{-5}$). Among these sites, 53.85% were also detected to have methylation-dependent activity, which is slightly, but not significantly higher than the proportion of methylation-dependent sites on the Illumina Methylation450K chip as a whole ($\log_2$(OR) [95% CI]=0.884 [-0.338, 2.130], p=0.16). Consequently, ELA-associated sites in this study were not more likely to exhibit methylation-dependent activity than chance. We speculate that regulatory enrichment in this data set is due to its focus on intermediately methylated CpG sites with substantial interindividual variance in DNA methylation levels, which tends to enrich for enhancer elements. Indeed, ELA-associated sites in this study were

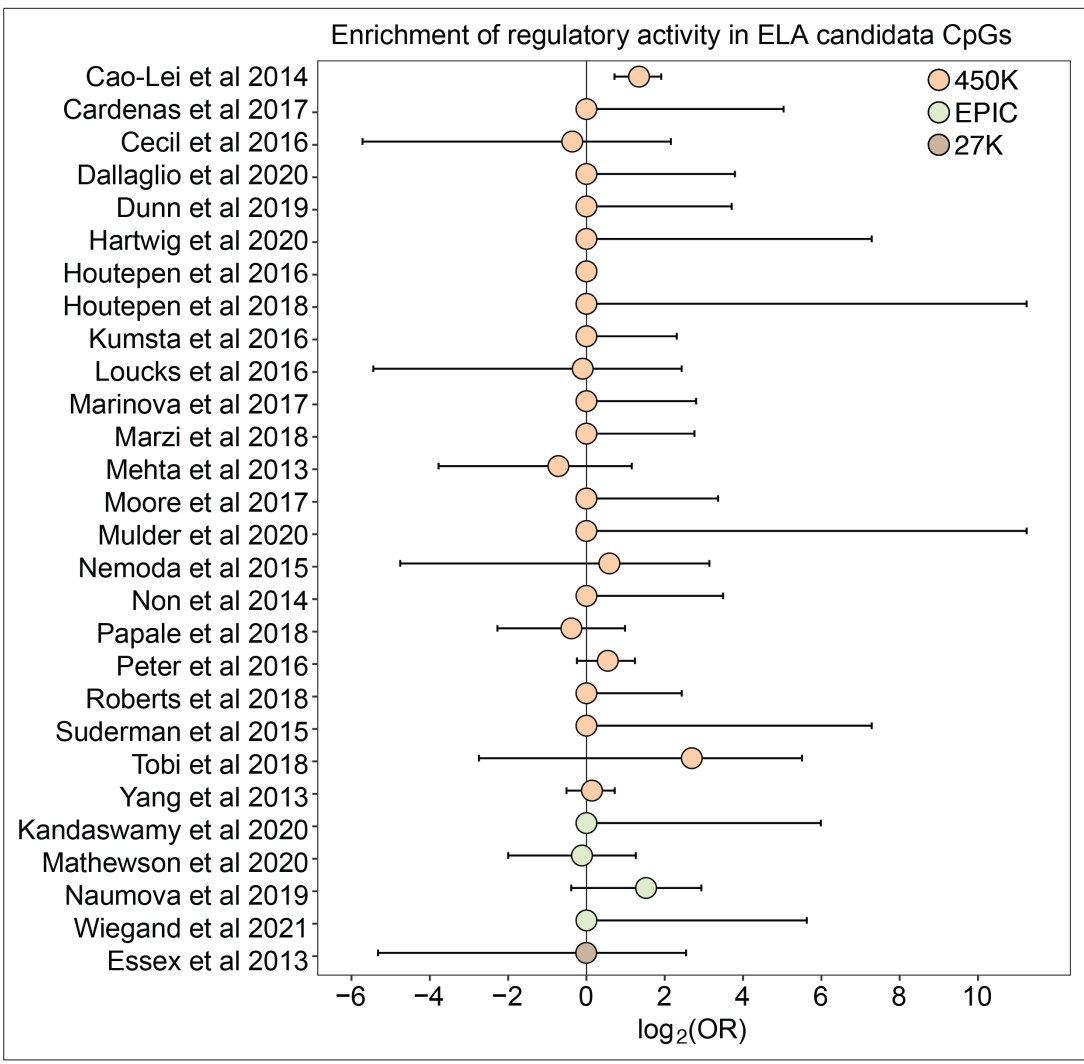

**Figure 4.** Early life adversity-associated CpG sites are not enriched for mSTARR-seq regulatory activity. $\log_2$-tranformed odds ratios from Fisher's Exact Tests for enrichment relative to the background set of sites on each array platform, for 27 studies of early life adversity-DNA methylation level correlations (see *Supplementary file 20* for full FET results). Whiskers show standard error. Only *Cao-Lei et al., 2014* shows significant enrichment for regulatory activity ($\log_2$(OR) [95% CI]=1.343 [0.715, 1.909], p=3.39 x $10^{-5}$), but these sites are not more likely to exhibit methylation-dependent activity than expected by chance ($\log_2$(OR) [95% CI]=0.884 [-0.338, 2.130], p=0.16). For details on the source tissue and measures of ELA, see *Figure 4—figure supplement 1*.

The online version of this article includes the following figure supplement(s) for figure 4:

**Figure supplement 1.** Summary of early-life adversity (ELA) studies.

more strongly enriched in enhancer regions than the union set of sites in other studies we investigated ($\log_2$(OR) [95% CI]=1.295 [0.543, 2.015], p=6.64 x $10^{-4}$).

## Discussion

Using mSTARR-seq, we assessed the functional role of DNA methylation across more than 99% of CpG sites assessed by two commonly used methods to measure methylation (the EPIC array and RRBS). Compared to our previous work, we identify a higher rate of methylation-dependence in the present data set (47.5% of analyzed windows in the current data set versus 10% of analyzed windows from *Lea et al., 2018* when we apply the current pipeline to analyze both datasets; *Supplementary files 3 and 6*). This difference likely stems from lower variance between replicates in the present study,

which increases power (*Figure 1—figure supplement 6*). Thus, our findings reveal that DNA methylation often significantly attenuates regulatory activity in K562 enhancer and promoter elements.

The results also support the idea that CpG sites identified by environment-DNA methylation association are a mixed bag. For example, despite the pervasive assumption in the literature and in popular science that early adversity causally impacts downstream outcomes through persistent epigenetic modification (e.g. *Dubois et al., 2019*), 98% of CpG sites associated with early life adversity in the literature fell in windows with no discernable regulatory potential in K562 cells, irrespective of their methylation status or cellular state. Our findings suggest that many ELA-associated sites may be better treated as passive biomarkers of exposure rather than links on the causal pathway between early life disadvantage and later life health outcomes. However, other cell types and cellular contexts should be tested to further evaluate this hypothesis (we used the erythroleukemic K562 cell line, as ELA-associated sites are most commonly assessed in blood). For instance, while we included the dexamethasone condition here because glucocorticoid dysregulation is commonly invoked in association with the response to early life adversity, the relationship between glucocorticoid signaling and early life adversity is complex (*Eisenberger and Cole, 2012*; *Koss and Gunnar, 2018*) and may not be well-modeled by acute glucocorticoid exposure. Given that the repertoire of poised and active enhancers often differs across cell types and cellular states, expanding experimental approaches like mSTARR-seq to other contexts can therefore serve a valuable role in prioritizing CpG sites identified in observational studies of differential methylation (e.g. CpG sites associated with disease).

Our results dovetail with work supporting the importance of CpG sites within enhancer elements in responding to environmental perturbations (*Pacis et al., 2015*; *Husquin et al., 2018*). Indeed, one of the most interesting findings from our analysis is the observation that DNA methylation-environment interactions are widespread in the human genome. Across thousands of genomic regions, both regulatory activity and methylation-dependent effects on regulatory activity were only detectable upon stimulation, consistent with a model in which DNA methylation contributes to epigenomic priming by modulating responsivity to environmental stimuli (*Kamada et al., 2018*; *Sun and Barreiro, 2020*). In support of this idea, IFNA-specific enhancers detected in mSTARR-seq are able to nonrandomly identify collections of loci in the genome where baseline DNA methylation also predicts the endogenous gene expression response to flu infection (*Figure 3B–C*). This relationship is easier to discern in within-individual comparisons across loci than in locus-by-locus analyses of interindividual variation. However, we were also able to pinpoint one specific gene, *ISG15*, in which baseline DNA methylation explains a large fraction of population variation in the gene expression response to flu (*Figure 3D–E*). Notably, the effect size for *ISG15* is very large, accounting for over a third of total variance in the fold-change gene expression response to flu. Unfortunately, a comparable in vivo dataset for dex challenge was unavailable to conduct parallel validation studies. However, our results overall suggest that mSTARR-seq could help identify additional in vivo DNA methylation-environment interactions, given sufficiently large sample sizes and data on the environmental exposures of interest.

Applying mSTARR-seq in additional cell types may therefore help resolve whether DNA methylation is responsible for exaggerated transcriptional responses to repeated challenges, as previously suggested for glucocorticoid exposure in hippocampal progenitor cells (*Provençal et al., 2020*), or whether differential methylation after pathogen infection contributes to the emerging paradigm of 'trained immunity' in innate immune cells (reviewed in *Fanucchi et al., 2021*). More generally, while our findings agree with the idea that many differences in DNA methylation—even extreme ones (~0% versus ~100%) like those tested here—are silent with respect to transcription factor binding and gene expression (e.g. *Maeder et al., 2013*; *Kreibich et al., 2023*), they also suggest that the functional importance of DNA methylation is likely to be underestimated without considering its interaction with the cellular environment.

## Materials and methods

**Key resources table**

| Reagent type (species) or resource | Designation | Source or reference | Identifiers | Additional information |
| --- | --- | --- | --- | --- |
| Commercial assay or kit | SeqCap EZ Prime Choice XL Probes | Roche | Cat # 08247528001 | |

*Continued on next page*

*Continued*

| Reagent type (species) or resource | Designation | Source or reference | Identifiers | Additional information |
|---|---|---|---|---|
| Commercial assay or kit | SeqCap EZ Reagent Kit Plus | Roche | Cat # 06953247001 | |
| Cell line (human) | K562 | ATCC | ATCC CCL-243 | |
| Cell line (human) | HepG2 | ATCC | Cat # HB-8065 | |
| Recombinant DNA reagent | pmSTARRseq1 | Addgene | Plasmid #96945 | |
| Peptide, recombinant protein | IFNA-2b | ThermoFisher Scientific | Cat # 111051 | |
| Chemical compound, drug | dexamethasone | Sigma-Aldrich | Cat # D4902 | |
| Other | GT115 strain chemically competent *E. coli* cells | Invivogen | ChemiComp GT115 | GT115 *E. coli* cells were initially purchased from Invivogen, and a custom electrocompetent version of the strain was prepared by Intact Genomics. |
| Other | GT115 strain electrically competent *E. coli* cells | Intact Genomics | | Custom order to grow Invivogen's ChemiComp GT115 strain and prepare cells for electroporation |
| Software, algorithm | R Project for Statistical Computing | R Project for Statistical Computing | RRID:SCR_001905 | |
| Software, algorithm | LIMMA | LIMMA | RRID:SCR_010943 | |
| Software, algorithm | HOMER | HOMER | RRID:SCR_010881 | |
| Software, algorithm | Trim Galore | Trim Galore | RRID:SCR_011847 | |
| Software, algorithm | cutadapt | cutadapt | RRID:SCR_011841 | |
| Software, algorithm | HTSeq | HTSeq | RRID:SCR_005514 | |
| Software, algorithm | bedtools | bedtools | RRID:SCR_006646 | |
| Software, algorithm | edgeR | edgeR | RRID:SCR_012802 | |
| Software, algorithm | sva package | sva package | RRID:SCR_012836 | |

## Cell lines

The K562 and HepG2 cell lines were obtained from ATCC, who performed cell line validation using short tandem repeat (STR) profiling and mycoplasma contamination testing (both lines tested negative).

## DNA capture and mSTARR-seq

We used the DNEasy Blood and Tissue Kit (Qiagen) to extract 5 µg DNA from the GM12878 lymphoblastoid cell line. We sheared the extracted DNA on a Covaris S2 with the following parameters: intensity = 3; duty cycle = 5%, cycles/burst = 200, treatment time = 40 s, temperature = 4 ° C; intensifier = yes. We then performed agarose gel size selection of ~600–700 bp DNA fragments followed by purification with the Qiaquick Gel Extraction Kit (QIAGEN). We note that we intentionally targeted longer fragments (~600–700 bp) than those targeted in *Lea et al., 2018* (~300–700 bp) because our previous work showed that longer fragments are more likely to drive regulatory activity (*Lea et al., 2018*).

We used Roche's NimbleDesign Software to design probes to capture the following genomic regions: (i) EPIC CpGs targeted by the Illumina Infinium MethylationEPIC microarray (862,831 CpGs); (ii) predicted CpG cut sites from an in silico Msp1 digest of the hg38 genome, followed by in silico size selection to 100–500 bp, which together simulates the first step in reduced representation bisulfite sequencing protocols (318,929 CpGs); (iii) 6.5 Mb centered around the *NR3C1* glucocorticoid receptor gene; and (iv) 100,099 loci included as a control for genomic background that were randomly distributed across the hg38 human genome, excluding centromeres, gaps, and Ns. Although the Y chromosome was included during the design of control loci, we excluded the Y chromosome in analyses, resulting in 99,999 control loci which fell into 98,967 unique 600 bp windows. Of these 98,967 windows, 94,679 contained CpG sites and 4,288 did not, reflecting the

sequence composition of the human genome as a whole. We note that a single 600 bp genomic window could simultaneously include EPIC CpGs, *Msp1* CpG cut sites, the *NR3C1* region, or control sites (i.e. assignment of windows to these compartments are not mutually exclusive; *Figure 1— figure supplement 1*).

We then captured target regions from the sheared GM12878 DNA using SeqCap EZ Prime Choice Probes (Roche), following the SeqCap EZ HyperCap protocol version 2.3. We performed two capture reactions. We used 6 cycles for the Pre-Capture LM-PCR program and 16 cycles for the Post-Capture LM-PCR program. For the Post-Capture LM-PCR reaction, we used the mSTARR_primerF2 and mstarr_ primerR2 primers from *Lea et al., 2018* to prepare the DNA for ligation into the mSTARR plasmid. The resulting captured DNA was used as input for the mSTARR-seq protocol (*Lea et al., 2018*), beginning at the mSTARR-seq published protocol step 'Generate linearized mSTARR backbone for large-scale cloning'.

We performed four and six replicate Gibson assemblies and transformations for the first and second capture reactions, respectively. We then sequenced each transformation replicate on a MiSeq to measure library diversity per replicate, with the following modifications to the published mSTARR-seq protocol: we used 25 µl Kapa HiFi HotStart ReadyMix instead of 25 µl NEB Q5 Hot start master mix, with the following cycle program: 98 C for 45 s, 12 cycles of 98 C for 15 s, 60 C for 30 s, and 72 C for 60 s, followed by a final extension at 72 C for 5 min. We measured library diversity of each transformation using the ENCODE Project's PCR bottleneck coefficient (PBC), which is the percentage of non-duplicate mapped reads from the total number of mapped reads (*Landt et al., 2012*; *Supplementary file 21*). We pooled the ten transformations by weighting their molarities according to the PBCs to create the final library. We then performed 8 re-transformations to expand the pooled library.

Replicate methyltransferase reactions and parallel mock methyltransferase reactions, in which the *M. SssI* enzyme was replaced with water, were performed in 500 µl reactions, cleaned with Ampure beads, and then pooled. The methylated DNA library and unmethylated DNA library were each transfected into 18 replicate T75 flasks, each containing 12 million K562s, with Lipofectamine 3000 (Thermo Fisher Scientific) following the manufacturer's instructions. Forty-two hours post-transfection, replicates were treated with 2000 U/mL IFNA2b (Thermo Fisher Scientific), 1 µM dex (Sigma-Aldrich), or vehicle control (media; six replicates per treatment, following *Lea et al., 2018*). Forty-eight hours post-transfection (6 hr post-treatment), cells were harvested for mSTARR-seq sequencing library generation. At harvest, $5 \times 10^5$ cells were aliquoted separately into Buffer RLT to measure endogenous RNA response to treatment, and 2 million cells were aliquoted for plasmid DNA extraction to measure input DNA in each replicate. mSTARR RNA-seq and DNA-seq libraries were generated following *Lea et al., 2018*. To measure endogenous gene expression from K562s in each condition, we extracted RNA from the separately aliquoted cells using the Qiagen RNEasy kit and prepared RNA-seq libraries using the NEBNext Ultra II RNA Library Prep Kit for Illumina.

## Comparison of mSTARR MD regulatory activity across cell types and experiments

To assess cell type-specificity of methylation-dependent regulatory activity, we transfected the mSTARR-seq library from *Lea et al., 2018*, comprised of 1:3 sheared genomic DNA:*MspI*-digested DNA, into the HepG2 cell line. For the HepG2 experiment, we performed one methyltransferase reaction and one sham methyltransferase reaction. Transfections were performed with Lipofectamine 3000 (Thermo Fisher Scientific) following the manufacturer's instructions, with reagent quantities scaled to the following per replicate: 6.9 million HepG2 cells, 46 µg of DNA, 138 µl of Lipofectamine 3000, and 180 µl of P3000 (transfection enhancer reagent). We performed all regulatory and MD regulatory analyses of HepG2 cells, as well as analyses of K562s from a previously published smaller mSTARR-seq experiment (*Lea et al., 2018*), following the same bioinformatics pipeline used for our main K562 experiment, described below. To compare against previously published STARR-seq data in A549 cells, we identified the union set of all regulatory peaks identified across six replicate STARR-seq experiments collected at time point 0 in *Johnson et al., 2018*. To test for significant overlap between the current K562 results and the other three data sets, we used Fisher's exact tests against a background set of windows tested in both the current K562 experiment and each of the other data sets.

## mSTARR-seq regulatory and MD regulatory analyses

mSTARR RNA-seq and DNA-seq libraries were sequenced on the Illumina NovaSeq platform as 100 basepair, paired-end reads (*Supplementary file 1*). Reads were trimmed with Trim Galore (version 0.6.4; *Krueger, 2019*) to remove basepairs at the ends of reads with Phred scores less than 20 and stretches of 2 or more basepairs that matched adapter sequences. Trimmed reads with a minimum length of 25 basepairs were retained. We mapped reads using *bwa* (version 0.7.12; *Li and Durbin, 2009*) using default settings. We retained read pairs that mapped to a single best location using the *samtools* (version 1.3.1; *Li et al., 2009*) package 'view' command with options -q 10 f 0 x2. We segmented the genome into 600 bp, non-overlapping windows, and used the bedtools (version 2.25.0; *Quinlan and Hall, 2010*) *coverage* function to count the number of RNA and DNA fragments overlapping each 600 bp window. We chose 600 bp for the genomic window size to accommodate our library's DNA fragment size (600–700 bp; *Supplementary file 1*), and because using smaller windows resulted in the identification of many adjacent windows as separate regulatory elements when they likely represent a single true enhancer (the median length of ENCODE-annotated enhancers is 600 bp).

For each of the three treatment conditions (baseline, IFNA, and dex), we reduced our dataset to windows for which at least three DNA samples in the methylated condition and three DNA samples in the unmethylated condition (i.e. 6 DNA samples total), and three RNA samples in *either* the methylated or unmethylated condition, had nonzero counts. One mSTARR RNA-seq sample in the baseline condition (sample ID L31250) was removed from further analysis because it had an unusually high proportion of zero counts in the testable windows; the corresponding paired DNA sample was therefore also removed prior to analysis (*Supplementary file 1*). Next, we retained only windows that showed high repeatability across DNA samples, following *Lea et al., 2018*. In brief, for each window we calculated the pairwise difference of read abundance for every pair of samples and created a distribution of all pairwise differences; windows were removed if at least 25% of pairs fell outside the central 90th percentile of the distribution (37,410 windows, 14.8% of the overall data set). On this reduced data set of testable windows, we performed voom normalization using the limma (version 3.44.3) *voomWithQualityWeights* function (*Smyth, 2005*; *Law et al., 2014*), with methylation status included as a covariate in the design. We then used the *limma* function *lmFit* to apply the following model for each window:

$$y_i = \mu + m_i\beta_1 + s_i\beta_2 * I\left(m = 0\right) + s_i\beta_3 * I\left(m = 1\right) + \varepsilon_i$$

where $y_i$ is the vector of normalized counts for n=24 samples (12 RNA and 12 DNA samples; 22 samples for the baseline condition); μ is the intercept; $m_i$ is methylation condition (0=unmethylated; 1=methylated) and $\beta_1$ its effect size; and I is an indicator variable in which $\beta_2$ and $\beta_3$ are the effects of sample type ($s_i$; 0=DNA; 1=RNA) in the unmethylated (m=0) and methylated conditions (m=1), respectively. $\varepsilon_i$ is the residual error.

To estimate the false discovery rate for identifying windows with regulatory activity (in either the methylated or unmethylated condition, or both), we compared the observed results to empirical null distributions generated using 100 permutations of RNA/DNA labels within each pair of samples following *Storey and Tibshirani, 2003*. Unlike in the real data, in which most windows show a strong bias towards more reads in the DNA condition (because most windows exhibit no regulatory activity, and therefore show many more input DNA reads than RNA reads), permuted data results in relatively balanced effect sizes. Using the overall distribution of p-values to construct the null therefore substantially inflates the number of significant windows, but specifically because most observed windows have systematically *less* activity in the RNA condition than the DNA condition, not more (the direction of interest). Thus, retaining all p-values to construct the null leads to highly miscalibrated false discovery rates because the distribution of observed values is skewed toward smaller values—because of windows with 'significantly' no regulatory activity—compared to the permuted data. To address this problem, for each permutation, we subsampled N windows with a positive sample type beta (corresponding to larger read counts in RNA samples versus DNA samples) in either the methylated condition, unmethylated condition, or both. Here, N is the number of windows with a positive sample type beta in the observed data (e.g. N=17,461 in the baseline dataset). The p-values from these windows were used to generate the empirical null. To define putative enhancer elements, we used an FDR cutoff of 1%.

To identify methylation-dependent enhancer activity, we focused on the windows that exhibited a positive sample type beta (i.e. exhibited more RNA reads than DNA reads) in either the methylated or unmethylated condition (or both) and tested for significant differences between $\beta_2$ and $\beta_3$ in the equation above (i.e. an interaction effect between RNA/DNA condition and the methylated/unmethylated condition). To estimate the false discovery rate for identifying methylation-dependent windows, we re-ran our main nested model on the same 100 sets of N windows each, in this case permuting methylation condition (unmethylated versus methylated) across sample pairs to generate the empirical null for the interaction effect of sample type (RNA versus DNA) * methylation condition (methylated or unmethylated). To define methylation-dependent regulatory elements, we again used an FDR cutoff of 1%.

To assess whether mSTARR RNA-seq and DNA-seq libraries were sequenced deeply enough to saturate detection of unique 600 bp windows included in the formal analyses, we used seqtk (https://github.com/lh3/seqtk, copy archived at *Li, 2024*) to randomly subset sequence reads from the raw fastq files for all RNA-seq or all DNA-seq replicates in the baseline dataset. We then applied to these data subsets the same filtering pipeline described above to generate a reduced data set of testable windows. The results of this rarefaction analysis (shown in *Figure 1—figure supplement 4*) show that our sequencing depth is sufficient to capture all analyzable windows based on both the DNA read filter and the RNA read filter.

## Enrichment of transcription factor binding motifs

To identify enrichment of potential transcription factor binding sites in the methylated condition, we used HOMER and motifs defined in the HOMER database (*Heinz et al., 2010*) and the baseline treatment dataset to compare motifs within regulatory regions in the methylated condition, relative to motifs across all regulatory regions.

To identify potential transcription factor binding sites in windows with condition-specific regulatory or methylation-dependent regulatory activity, we first identified regions showing regulatory activity uniquely in the dex or IFNA conditions relative to both other conditions (based on an FDR of 1% to define regulatory activity). We used HOMER and motifs defined in the HOMER database to test for enrichment of transcription factor binding motifs within windows showing condition-specific regulatory activity relative to all windows tested for regulatory activity in that condition. We set a threshold of Bonferroni-corrected p-value <0.01 to identify significantly enriched binding motifs.

## Endogenous gene expression response of K562s to dex and IFNA

Endogenous RNA-seq libraries from K562 cells challenged with dex or IFNA were sequenced on an Illumina NovaSeq 6000 S1 flow cell as 100 base pair, paired-end reads. Reads were trimmed with Trim Galore (Trim Galore version 0.6.4_dev; Cutadapt version 2.3) to remove basepairs with a Phred score less than 20, and end sequences that matched at least two basepairs of the adapter sequence. Only trimmed reads longer than 25 basepairs were retained. We used the STAR package (version 2.5.0; *Dobin et al., 2013*) two-pass mapping to map the filtered reads to the hg38 genome. We retained uniquely mapped reads by filtering the output SAM file to keep reads with MAPQ = 255. We then used *htseq* (version 0.6.0; *Anders et al., 2015*) to quantify read counts per gene. We only retained genes that had TPM >3 in at least three of six samples in at least one of the three conditions (baseline, IFNA, or dex). Only protein-coding genes were retained for final analysis, resulting in a total of 10,676 testable genes.

We performed differential expression analysis separately for IFNA and dex, by subsetting the data to the IFNA and baseline samples, or subsetting the data to the dex and baseline samples. For each subset, we *voom* normalized the data (*Law et al., 2014*), and used the *limma* (*Smyth, 2005*) function *lmFit* to model the normalized gene expression as a function of treatment, controlling for the percent of reads mapping to annotated features (to account for technical variation in the efficiency of RNA purification during library prep) and methylation condition for the associated sample. We calculated the FDR using the q-value approach of *Storey and Tibshirani, 2003*, based on an empirical null distribution that was derived from 100 permutations of treatment condition in the model.

To test whether genes were more responsive to IFNA stimulation if they were predicted targets of IFNA-specific ISRE enhancers, we first used the bedtools *intersect* function to identify the union set of ChIP-seq peaks for STAT1 and STAT2 (two of the three components of the ISGF3 transcription factor

that binds ISRE motifs; ENCODE accession numbers ENCFF478XGE and ENCFF394KTR; ChIP-Seq data were not available for the third ISGF3 component, IRF9). We then reduced these regions to those showing significant, IFNA condition-specific regulatory activity in the mSTARR-seq dataset (FDR <1%). We used K562 enhancer-gene links from EnhancerAtlas 2.0 (*Gao et al., 2016*) to link the resulting IFNA-specific ISRE enhancers to their target genes. Finally, we performed a two-sided, unpaired t-test to compare the endogenous expression responses to IFNA for genes associated with putative IFNA-specific ISRE enhancers relative to genes that are not associated with IFNA-specific ISRE enhancers.

## Endogenous gene expression and methylation in human macrophages

To assess effects of DNA methylation-environment interactions on gene expression in vivo, we evaluated endogenous methylation and gene expression from matched whole genome bisulfite sequencing (WGBS) and RNA-seq data collected from human monocyte-derived macrophages (n=35 donors), with and without infection with the influenza A virus (IAV; *Aracena et al., 2024*). Unsmoothed methylation counts were obtained for 19,492,906 loci in both non-infected and IAV-infected samples (total n=70) as described (*Aracena et al., 2024*). We filtered loci to require coverage of ≥4 sequence reads in at least half of the non-infected or IAV-infected samples. In the RNA-seq dataset for the same 35 individuals, we excluded any genes that did not have an average RPKM >2 in non-infected or IAV-infected samples. This resulted in a total of 19,041,420 CpG sites and 14,122 genes used in downstream analyses.

We calculated normalization factors using calcNormFactors in edgeR (v 3.28.1; *Robinson et al., 2010*) to scale the raw library sizes. We then used the voom function in limma (v 3.42.2; *Smyth, 2005*; *Ritchie et al., 2015*) to apply the normalization factors, estimate the mean-variance relationship, and convert raw read counts to log(CPM) values. Sequencing batches were regressed out using ComBat from the sva Bioconductor package (v 3.34.0; *Leek et al., 2012*), which fit a model that also included age (mean-centered) and admixture estimates. We subsequently regressed out age effects using limma. Individual-wise fold-change (FC) matrices were built by subtracting non-infected counts from IAV-infected counts for each individual using weights calculated as in *Harrison et al., 2019*; *Aracena et al., 2024*.

For comparison of the mSTARR-seq dataset to the dataset from *Aracena et al., 2024*, GrCh38/hg38 coordinates were lifted over to GRCh37/hg19 using the UCSC liftOver tool. We required a 0.95 minimum ratio of bases that remap, excluded loci that output multiple regions, set the minimum hit size in query to 0, and set the minimum chain size in the target to 0. The GRCh37/hg19 coordinates were used in the following analyses.

We first sought to assess, within each of the 35 individuals, the correlation between mSTARR-seq enhancer methylation levels at baseline (i.e. in non-infected cells) and transcriptional responses of their nearest genes to IAV. To find overlaps between enhancer regions and CpG loci, we used the *bedtools* intersect function (v2.29.1) with the 'left outer join' option. For each 600 bp mSTARR-seq enhancer region in each individual, we calculated the mean methylation level across all overlapping CpGs. Each enhancer was linked to its nearest gene, with a maximum distance of 100 kb, as described above. If an enhancer had multiple linked genes (e.g. an enhancer that overlapped more than one gene), we took the mean transcriptional response of all linked genes. For each individual, we calculated the Pearson's correlation coefficient (r) between the DNA methylation levels within the mSTARR-seq-defined enhancer windows, in non-infected cells, and transcriptional responses of their linked genes to IAV infection. We also investigated the correlation between mean mSTARR-seq enhancer window methylation and gene expression *within* non-infected and flu-infected samples separately.

Finally, for each mSTARR-seq enhancer-gene pair, we sought to test the extent to which average methylation level in the enhancer in non-infected samples explained the gene's transcriptional response to IAV, across individuals. Here, we calculated the average CpG methylation level for each 600 bp enhancer, after excluding CpGs with methylation variance less than 0.01 across individuals. Thus, enhancers that did not contain any CpGs with appreciable interindividual variation in DNA methylation levels were excluded from the analysis. This filtering step resulted in 1,382 enhancer-gene pairs for this analysis. For each enhancer-gene pair, we calculated the Pearson's correlation coefficient (and $R^2$) between the average methylation level of the enhancer and the transcriptional response of the linked gene. p-values were corrected for multiple hypothesis testing using the q-value method in R (*Storey and Tibshirani, 2003*).

## CpG methylation associated with early life adversity

We performed a literature search to identify studies for which CpG methylation differences have been linked to adverse conditions during early life. We considered all journal articles that contained the words 'early life adversity' and 'Infinium' on https://scholar.google.com on May 10, 2021, which produced 269 results. We required that the study evaluated CpG methylation using an Illumina Infinium array (27K, 450K, or EPIC), and that the article provided Infinium CpG probe IDs or CpG genomic coordinates of candidate CpGs. Articles that reported no significant CpG sites, but still reported 'top' CpG sites, were retained. Articles that performed analyses to identify differentially methylated regions (DMRs; as opposed to CpG site-by-site analysis), and then reported individual CpG sites within the candidate DMRs, were retained. We considered early life adversity to encompass both social and nonsocial sources of environmental adversity (e.g. exposure to severe weather), any time from prenatal development to 18 years of age. We did not impose criteria for cell type or subject age at time of sampling. See *Figure 4—figure supplement 1* and *Supplementary file 20* for a summary of the resulting studies.

For each study dataset, we performed a Fisher's exact test to test whether ELA candidate sites, relative to the total set of CpG sites on the Illumina array used in that study (450K, EPIC, or 27K array), were enriched in the mSTARR-seq regulatory windows we identified in the baseline, dex, or IFNA condition.

## Acknowledgements

We thank BJ Nielsen, Tawni Voyles, and Tina Del Carpio for their contributions to data generation, Terrie Moffitt and the Reddy lab at Duke University for project feedback, Graham Johnson for sharing the A549 data, Courtney Karner for use of equipment, Xiang Zhou for constructive advice on statistical modeling, and two anonymous reviewers for constructive feedback on an earlier version of this manuscript. This study was supported by the National Institutes of Health (R01HD088558 to JT and F32HD095616 to RAJ), the Canadian Institutes for Advanced Research Child Brain and Development Program, a Sloan Foundation Early Career Research Fellowship to JT, and a Foerster-Bernstein Postdoctoral Fellowship to RAJ. High-performance computing resources were supported by the North Carolina Biotechnology Center (Grant Number 2016-IDG-1013 and 2020-IIG-2109).

## Additional information

### Competing interests

Jenny Tung: Reviewing editor, eLife. The other authors declare that no competing interests exist.

### Funding

| Funder | Grant reference number | Author |
|---|---|---|
| National Institutes of Health | R01HD088558 | Jenny Tung |
| National Institutes of Health | F32HD095616 | Rachel A Johnston |
| Canadian Institute for Advanced Research | Child Brain and Development Program | Jenny Tung |
| Sloan Foundation | Early Career Research Fellowship | Jenny Tung |
| Foerster-Bernstein Postdoctoral Fellowship | | Rachel A Johnston |

The funders had no role in study design, data collection and interpretation, or the decision to submit the work for publication. Open access funding provided by Max Planck Society.

## Author contributions
Rachel A Johnston, Conceptualization, Data curation, Formal analysis, Funding acquisition, Validation, Investigation, Visualization, Methodology, Writing – original draft, Writing – review and editing; Katherine A Aracena, Formal analysis, Visualization, Writing – original draft, Writing – review and editing; Luis B Barreiro, Supervision, Writing – review and editing; Amanda J Lea, Methodology, Writing – review and editing; Jenny Tung, Conceptualization, Formal analysis, Supervision, Funding acquisition, Investigation, Writing – original draft, Writing – review and editing

## Author ORCIDs
Rachel A Johnston ⬤ https://orcid.org/0000-0002-8965-1162
Katherine A Aracena ⬤ http://orcid.org/0000-0002-8830-904X
Amanda J Lea ⬤ https://orcid.org/0000-0002-8827-2750
Jenny Tung ⬤ https://orcid.org/0000-0003-0416-2958

Reviewer #1 (Public Review): https://doi.org/10.7554/eLife.89371.3.sa1
Reviewer #2 (Public Review): https://doi.org/10.7554/eLife.89371.3.sa2
Author Response https://doi.org/10.7554/eLife.89371.3.sa3

# Additional files

## Supplementary files
• Supplementary file 1. Table summarizing the K562 mSTARR-seq DNA and RNA libraries generated in this study.

• Supplementary file 2. Table providing sequencing depth and CpG composition statistics for the target regions (EPIC array, RRBS, *NR3C1*, and control) in the baseline dataset.

• Supplementary file 3. Table of results from the model testing for methylation-dependent regulatory activity in K562 cells at baseline.

• Supplementary file 4. Table of enrichment results of mSTARR-seq regulatory regions for ENCODE-defined chromosome states.

• Supplementary file 5. Table providing sequencing depth and CpG composition statistics for the baseline dataset and *Lea et al., 2018* dataset.

• Supplementary file 6. Table of results from the model testing for methylation-dependent regulatory activity in K562 cells at baseline condition using the dataset from *Lea et al., 2018*.

• Supplementary file 7. Table of results from the model testing for methylation-dependent regulatory activity in HepG2 cells at baseline.

• Supplementary file 8. Table of enrichment results of transcription factor binding motifs for regulatory windows with higher activity in the methylated relative to the sham condition.

• Supplementary file 9. Table providing the number of 600 bp windows passing each filter for the baseline, IFNA, dex, and HepG2 datasets.

• Supplementary file 10. Table of results from the model testing for methylation-dependent regulatory activity in K562 cells stimulated with IFNA.

• Supplementary file 11. Table of results from the model testing for methylation-dependent regulatory activity in K562 cells stimulated with dex.

• Supplementary file 12. Table of enrichment results of transcription factor binding motifs for regulatory windows specific to the IFNA treatment (<1% FDR in IFNA and >1% FDR in baseline and Dex).

• Supplementary file 13. Table of enrichment results of transcription factor binding motifs for regulatory windows specific to the Dex treatment (<1% FDR in Dex and >1% FDR in baseline and IFNA).

• Supplementary file 14. Table of enrichment results of transcription factor binding motifs for regulatory windows specific to the IFNA treatment (<1% FDR in IFNA and >10% FDR in baseline and Dex).

• Supplementary file 15. Table of enrichment results of transcription factor binding motifs for regulatory windows specific to the Dex treatment (<1% FDR in Dex and >10% FDR in baseline and IFNA).

- Supplementary file 16. Table of results from the model testing for the K562 endogenous gene expression response to IFNA.

- Supplementary file 17. Table of Pearson's correlation results, within individuals, between mSTARR-seq enhancer DNA methylation in non-infected macrophages and transcriptional response of nearest genes upon IAV infection.

- Supplementary file 18. Table of Pearson's correlation results, within individuals, between mSTARR-seq enhancer DNA methylation in non-infected macrophages and gene expression level in non-infected or IAV-infected macrophages.

- Supplementary file 19. Table of Pearson's correlation results, across individuals, between mSTARR-seq enhancer DNA methylation in non-infected macrophages and transcriptional response of nearest genes upon IAV infection.

- Supplementary file 20. Table of enrichment results of regulatory activity in early-life adversity associated CpGs relative to regulatory activity in all CpGs on the associated Illumina array.

- Supplementary file 21. Library diversity of replicate transformations using the ENCODE Project's PCR bottleneck coefficient (PBC).

- MDAR checklist

## Data availability

mSTARR-seq RNA and DNA sequencing data generated in this study for K562 and HepG2 cells are available in the NCBI Sequence Read Archive (SRA; BioProject accession number PRJNA922490). K562 endogenous RNA-seq data are available in the NCBI Gene Expression Omnibus (GEO series accession GSE222643). The previously published data from *Lea et al., 2018* used in this study are available in NCBI's SRA accession number SRP120556. The previously published A549 data from *Johnson et al., 2018* were kindly provided by Graham Johnson. The sequencing data for the macrophage RNA-seq and WGBS datasets are available at the European Genome-Phenome Archive (EGA) under accession numbers EGAD00001008422 and EGAD00001008359, respectively. The Zenodo repository at https://zenodo.org/record/7949036#.ZGZ5UnbMJq9 provides the code used for analyses, as well as track files (compatible with the UCSC genome browser) for mSTARR-seq hg38 windows with regulatory activity, and windows with methylation-dependent regulatory activity, in all three conditions (baseline, IFNA, and dex). The detailed step-by-step mSTARR-seq protocol is available online at http://www.tung-lab.org/protocols-and-software.html. The mSTARR-seq plasmid, pmSTARRseq1 is available through AddGene.

The following datasets were generated:

| Author(s) | Year | Dataset title | Dataset URL | Database and Identifier |
|---|---|---|---|---|
| Johnston RA, Aracena KA, Barreiro LB, Lea AJ, Tung J | 2023 | DNA methylation-environment interactions in the human genome | https://www.ncbi.nlm.nih.gov/sra/?term=PRJNA922490 | NCBI Sequence Read Archive, PRJNA922490 |
| Johnston RA, Aracena KA, Barreiro LB, Lea AJ, Tung J | 2023 | DNA methylation-environment interactions in the human genome | https://www.ncbi.nlm.nih.gov/geo/query/acc.cgi?acc=GSE222643 | NCBI Gene Expression Omnibus, GSE222643 |
| Rachel AJ, Katherine AA, Luis BB, Amanda JL, Jenny T | 2023 | DNA methylation-environment interactions in the human genome | https://doi.org/10.5281/zenodo.7949035 | Zenodo, 10.5281/zenodo.7949035 |

The following previously published datasets were used:

| Author(s) | Year | Dataset title | Dataset URL | Database and Identifier |
|---|---|---|---|---|
| Lea AJ | 2018 | Genome-wide quantification of the effects of DNA methylation on human gene regulation | https://www.ncbi.nlm.nih.gov/sra/?term=SRP120556 | NCBI Sequence Read Archive, SRP120556 |

*Continued on next page*

*Continued*

| Author(s) | Year | Dataset title | Dataset URL | Database and Identifier |
|---|---|---|---|---|
| Groza C, Chen X, Pacis A, Simon MM, Pramatarova A, Aracena KA, Pastinen T, Barreiro LB, Bourque G | 2023 | Genetic drivers of epigenetic and transcriptional variation of human immune responses to infection (RNA-seq, ATAC-seq and ChIPmentation) | https://ega-archive.org/datasets/EGAD00001008422 | European Genome-Phenome Archive, EGAD00001008422 |
| Groza C, Chen X, Pacis A, Simon MM, Pramatarova A, Aracena KA, Pastinen T, Barreiro LB, Bourque G | 2023 | Genetic drivers of epigenetic and transcriptional variation of human immune responses to infection (RNA-seq, ATAC-seq and ChIPmentation) | https://ega-archive.org/datasets/EGAD00001008359 | European Genome-Phenome Archive, EGAD00001008359 |

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
