## [Editor Report · eLife assessment]

This **important** paper uses a genome-wide, massively parallel reporter assay to determine how CpG methylation affects regulatory sequences that control the expression of human genes. The authors provide **compelling** evidence that methylation not only influences baseline activity of regulatory sequences but also the magnitude of acute responses to environmental stimuli. The findings are of broad interest, and the extensive data set will likely become a key resource for the community.

---

## [Referee Report · Reviewer #1 (Public Review)]

In this manuscript, the authors explore the effects of DNA methylation on the strength of regulatory activity using massively parallel reporter assays in cell lines on a genome-wide level. This is a follow-up of their first paper from 2018 that describes this method for the first time. In addition to adding more in depth information on sequences that are explored by many researchers using two main methods, reduced bisulfite sequencing and sites represented on the Illumina EPIC array, they now show also that DNA methylation can influence changes in regulatory activity following a specific stimulation, even in absence of baseline effects of DNA methylation on activity. In this manuscript, the authors explore the effects of DNA methylation on the response to Interferon alpha (INFA) and a glucocorticoid receptor agonist (dexamethasone). The author validate their baseline findings using additional datasets, including RNAseq data and show convergences across two cell lines. The authors then map the methylation x environmental challenge (IFNA and dex) sequences identified in vitro to explore whether their methylation status is also predictive of regulatory activity in vivo. This is very convincingly shown for INFA response sequences, where baseline methylation is predictive of the transcriptional response to flu infection in human macrophages, an infection that triggers the INF pathways. The extension of the functional validity of the dex-response altering sequences is less convincing. Sequences altering the response to glucocorticoids, however, were not enriched in DNA methylation sites associated with exposure to early adversity which the authors interpret that "they are not links on the causal pathway between early life disadvantage and later life health outcomes, but rather passive biomarkers. However, this approach does not seem an optimal model to explore this relationship in vivo. This is because exposure to early adversity and its consequences is not directly correlated with glucocorticoid release and changes in DNA methylation levels following early adversity could be related to many physiological mechanisms, and overall, large datasets and meta-analyses do not show robust associations of exposure to early adversity and DNA methylation changes. Here other datasets, such as from Cushing patients maybe of more interest.

***

After revision, the authors have now discussed this issue carefully, so that this point is addressed.

***

Overall, the authors provide a great resource of DNA methylation sensitive enhancers that can now be used for functional interpretation of large scale datasets (that are widely generated in the research community), given the focus on sites included in RBSS and the Illumina EPIC array. In addition, their data lends support that difference in DNA methylation can alter responses to environmental stimuli and thus of the possibility that environmental exposures that alter DNS methylation can also alter subsequent response to this exposure, in line with the theory of epigenetic embedding of prior stimuli/experiences. The conclusions related to the early adversity data should be reconsidered in light of the comments above.

---

## [Referee Report · Reviewer #2 (Public Review)]

This work presents a remarkably extensive set of experiments, assaying the interaction between methylation and expression across most CpG positions in the genome in two cell types. To this end, the authors use mSTARR-seq, a high-throughput method, which they have previously developed, where sequences are tested for their regulatory activity in two conditions (methylated and unmethylated) using a reporter gene. The authors use these data to study two aspects of DNA methylation: 1. Its effect on expression, and 2. Its interaction with the environment. Overall, they identify a small number of 600 bp windows that show regulatory potential, and a relatively large fraction of these show an effect of methylation on expression. In addition, the authors find regions exhibiting methylation-dependent response to two environmental stimuli (interferon alpha and glucocorticoid dexamethasone).

The questions the authors address represent some of the most central in functional genomics, and the method utilized is currently the best method to do so. The scope of this study is very impressive and I am certain that these data will become an important resource for the community. The authors are also able to report several important findings, including that pre-existing DNA methylation patterns can influence the response to subsequent environmental exposures.

---

## [Author Response]

The following is the authors’ response to the original reviews.

**Reviewer #1 (Public Review):**
In this manuscript, the authors explore the effects of DNA methylation on the strength of regulatory activity using massively parallel reporter assays in cell lines on a genome-wide level. This is a follow-up of their first paper from 2018 that describes this method for the first time. In addition to adding more indepth information on sequences that are explored by many researchers using two main methods, reduced bisulfite sequencing and sites represented on the Illumina EPIC array, they now show also that DNA methylation can influence changes in regulatory activity following a specific stimulation, even in absence of baseline effects of DNA methylation on activity. In this manuscript, the authors explore the effects of DNA methylation on the response to Interferon alpha (INFA) and a glucocorticoid receptor agonist (dexamethasone). The authors validate their baseline findings using additional datasets, including RNAseq data, and show convergences across two cell lines. The authors then map the methylation x environmental challenge (IFNA and dex) sequences identified in vitro to explore whether their methylation status is also predictive of regulatory activity in vivo. This is very convincingly shown for INFA response sequences, where baseline methylation is predictive of the transcriptional response to flu infection in human macrophages, an infection that triggers the INF pathways.

Thank you for your strong assessment of our work!

The extension of the functional validity of the dex-response altering sequences is less convincing.

We agree. We note that genes close to dex-specific mSTARR-seq enhancers tend to be more strongly upregulated after dex stimulation than those near shared enhancers, which parallels our results for IFNA (lines 341-344). However, there is unfortunately no comparable data set to the human flu data set (i.e., with population-based whole genome-bisulfite sequencing data before and after dex challenge), so we could not perform a parallel in vivo validation step. We have added this caveat to the revised manuscript (lines 555-557).

Sequences altering the response to glucocorticoids, however, were not enriched in DNA methylation sites associated with exposure to early adversity. The authors interpret that "they are not links on the causal pathway between early life disadvantage and later life health outcomes, but rather passive biomarkers". However, this approach does not seem an optimal model to explore this relationship in vivo. This is because exposure to early adversity and its consequences is not directly correlated with glucocorticoid release and changes in DNA methylation levels following early adversity could be related to many physiological mechanisms, and overall, large datasets and meta-analyses do not show robust associations of exposure to early adversity and DNA methylation changes. Here, other datasets, such as from Cushing patients may be of more interest.

Thank you for making these important points. We have expanded the set of caveats regarding the lack of enrichment of early adversity-reported sites in the mSTARR-data set (lines 527-533). Specifically, we note that the relationship between early adversity and glucocorticoid physiology is complex (e.g., Eisenberger and Cole, 2012; Koss and Gunnar, 2018) and that dex challenge models one aspect of glucocorticoid signaling but not others (e.g., glucocorticoid resistance). Nevertheless, we also see little evidence for enrichment of early adversity-associated sites in the mSTARR data set at baseline, independently of the dex challenge experiment (lines 483-485; Figure 4).

We also agree that large data sets (e.g., Houtepen et al., 2018; Marzi et al., 2018) and reviews (e.g., Cecil et al., 2020) of early adversity and DNA methylation in humans show limited evidence of associations between early adversity and DNA methylation levels. However, the idea that early adversity impacts downstream outcomes remains pervasive in the literature and popular science (see Dubois et al., 2019), which we believe makes tests like ours important to pursue. We also hope that our data set (and others generated through these methods) will be useful in interpreting other settings in which differential methylation is of interest as well—in line with your comment below. We have clarified both of these points in the revised manuscript (lines 520-522; 536-539).

Overall, the authors provide a great resource of DNA methylation-sensitive enhancers that can now be used for functional interpretation of large-scale datasets (that are widely generated in the research community), given the focus on sites included in RBSS and the Illumina EPIC array. In addition, their data lends support that differences in DNA methylation can alter responses to environmental stimuli and thus of the possibility that environmental exposures that alter DNS methylation can also alter the subsequent response to this exposure, in line with the theory of epigenetic embedding of prior stimuli/experiences. The conclusions related to the early adversity data should be reconsidered in light of the comments above.

Thank you! And yes, we have revised our discussion of early life adversity effects as discussed above.

**Reviewer #1 (Recommendations For The Authors):**
While the paper has a lot of strengths and provides new insight into the epigenomic regulation of enhancers as well as being a great resource, there are some aspects that would benefit from clarification.a. It would be great to have a clearer description of how many sequences are actually passing QC in the different datasets and what the respective overlaps are in bps or 600bp windows. Now often only % are given. Maybe a table/Venn diagram for overview of the experiments and assessed sequences would help here. This concern the different experiments in the K652, A549, and Hep2G cell lines, including stimulations.

We now provide a supplementary figure and supplementary table providing, for each dataset, the number of 600 bp windows passing each filter (Figure 2-figure supplement 1; Supplementary File 9), as well as a supplementary figure providing an upset plot to show the number of assessed sequences shared across the experiments (Figure 2-figure supplement 2).

b. It would also be helpful to have a brief description of the main differences in assessed sequences and their coverage of the old (2018) and new libraries in the main text to be able better interpret the validation experiments.

We now provide information on the following characteristics for the 2018 data set versus the data set presented for the first time here: mean (± SD) number of CpGs per fragment; mean (± SD) DNA sequencing depth; and mean (± SD) RNA sequencing depth (lines 169-170 provide values for the new data set; in line 194, we reference Supplementary File 5, which provides the same values for the old data set). Notably, the coverage characteristics of analyzed windows in both data sets are quite high (mean DNA-seq read coverage = 94x and mean RNA-seq read coverage = 165x in the new data set at baseline; mean DNA-seq read coverage = 22x and mean RNA-seq read coverage = 54x in Lea et al. 2018).

c. Statements of genome-wide analyses in the abstract and discussion should be a bit tempered, as quite a number of tested sites do not pass QC and do not enter the analysis. From the results it seems like from over 4.5 million sequences, only 200,000 are entering the analysis.

The reason why many of the windows are not taken forward into our formal modeling analysis is that they fail our filter for RNA reads because they are never (or almost never) transcribed—not because there was no opportunity for transcription (i.e., the region was indeed assessed in our DNA library, and did not show output transcription, as now shown in Figure 2-figure supplement 1). We have added a rarefaction analysis (lines 715-722 in Materials and Methods) of the DNA fragment reads to the revised manuscript which supports this point. Specifically, it shows that we are saturated for representation of unique genomic windows (i.e., we are above the stage in the curve where the proportion of active windows would increase with more sequencing: Figure 1figure supplement 4). Similarly, a parallel rarefaction curve for the mSTARR-seq RNA-seq data (Figure 1-figure supplement 4) shows that we would gain minimal additional evidence for regulatory activity with more sequencing depth. We now reference these analyses in revised lines 179-184 and point to the supporting figure in line 182.

In other words, our analysis is truly genome-wide, based on the input sequences we tested. Most of the genome just doesn’t have regulatory activity in this assay, despite the potential for it to be detected given that the relevant sequences were successfully transfected into the cells.

d. Could the authors comment on the validity of the analysis if only one copy is present (cut-off for QC)?

We think this question reflects a misunderstanding of our filtering criteria due to lack of clarity on our part, which we have modified in the revision. We now specify that the mean DNA-seq sequencing depth per sample for the windows we subjected to formal modeling was quite high:

93.91 ± 10.09 SD (range = 74.5 – 113.5x) (see revised lines 169-170). In other words, we never analyze windows in which there is scant evidence that plasmids containing the relevant sequence were successfully transfected (lines 170-172).

Our minimal RNA-seq criteria require non-zero counts in at least 3 replicate samples within either the methylated condition or the unmethylated condition, or both (lines 166-168). Because we know that multiple plasmids containing the corresponding sequence are present for all of these windows—even those that just cross the minimal RNA-seq filtering threshold—we believe our results provide valid evidence that all analyzed windows present the opportunity to detect enhancer activity, but many do not act as enhancers (i.e., do not result in transcribed RNA). Notably, we observe a negligible correlation between DNA sequencing depth for a fragment, among analyzed windows, and mSTARR-seq enhancer activity (R2 = 0.029; now reported in lines 183-184). We also now report reproducibility between replicates, in which all replicate pairs have r > 0.89, on par with previously published STARR-seq datasets (e.g., Klein et al., 2020; Figure 1-figure supplement 6, pointed to in line 193).

e. While the authors state that almost all of the control sequences contain CpGs sites, could the authors also give information on the total number of CpG sites in the different subsets? Was the number of CpGs in a 600 bp window related to the effects of DNA methylation on enhancer activity?

We now provide the number of CpG sites per window in the different subsets in lines 282-284. As expected, they are higher for EPIC array sites and for RRBS sites because the EPIC array is biased towards CpG-rich promoter regions, and the enzyme typically used in the starting step of RRBS digests DNA at CpG motifs (but control sequences still contain an average of ~13 CpG sites per fragment). We also now model the magnitude of the effects of DNA methylation on regulatory activity as a function of number of CpG sites within the 600 bp windows. Consistent with our previous work in Lea et al., 2018, we find that mSTARR-seq enhancers with more CpGs tend to be repressed by DNA methylation (now reported in lines 216-219 and Figure 1figure supplement 11).

f. In the discussion, a statement on the underrepresented regions, likely regulatory elements with lower CG content, that nonetheless can be highly relevant for gene regulation would be important to put the data in perspective.

Thanks for this suggestion. We agree that regulatory regions, independent of CpG methylation, can be highly relevant, and now clarify in the main text that the “unmethylated” condition of mSTARR-seq is essentially akin to a conventional STARR-seq experiment, in that it assesses regulatory activity regardless of CpG content or methylation status (lines 128-130).

Consequently, our study is well-designed to detect enhancer-like activity, even in windows with low GC content. We now show with additional analyses that we generated adequate DNA-seq coverage on the transfected plasmids to analyze 90.2% of the human genome, including target regions with no or low CpG content (lines 148-149; 153-156; Supplementary file 2). As noted above, we also now clarify that regions dropped out of our formal analysis because we had little to no evidence that any transcription was occurring at those loci, not because sequences for those regions were not successfully transfected into cells (see responses above and new Figure 1-figure supplement 4 and Figure 2-figure supplement 1).

g. To control for differences in methylation of the two libraries, the authors sequence a single CpGs in the vector. Could the authors look at DNA methylation of the 600 bp windows at the end of the experiment, could DNA methylation of these windows be differently affected according to sequence? 48 hours could be enough for de-methylation or re-methylation.

We agree that variation in demethylation or remethylation depending on fragment sequence is possible. We now state this caveat in the main text (lines 158-159), and specify that genomic coverage of our bisulfite sequencing data across replicates are (unfortunately) too variable to perform reliable site-by-site analysis of DNA methylation levels before and after the 48 hour experiment (lines 1182-1185). Instead, we focus on a CpG site contained in the adapter sequence (and thus included in all plasmids) to generate a global estimate of per replicate methylation levels. We also now note that any de-methylation or re-methylation would reduce our power to detect methylation-dependent activity, rather than leading to false positives (lines 163-165).

h. The section on the method for correction for multiple testing should be more detailed as it is very difficult to follow. Why were only 100 permutations used, the empirical p-value could then only be <0.01? The description of a subsample of the N windows with positive Betas is unclear, should the permutation not include the actual values and thus all windows - or were the no negative Betas? Was FDR accounting for all elements and pairs?

We have now expanded the text in the Materials and Methods section to clarify the FDR calculation (lines 691, 695-699, 702, 706). We clarify that the 100 permutations were used to generate a null distribution of p-values for the data set (e.g., 100 x 17,461 p-values for the baseline data set), which we used to derive a false discovery rate. Because we base our evidence on FDRs, we therefore compare the distribution of observed p-values to the distribution of pvalues obtained via permutation; we do not calculate individual p-values by comparing an observed test statistic against the test statistics for permuted data for that individual window.

We compare the data to permutations with only positive betas because in the observed data, we observe many negative betas. These correspond to windows which have no regulatory activity (i.e., they have many more input DNA reads than RNA-seq reads) and thus have very small pvalues in a model testing for DNA-RNA abundance differences. However, we are interested in controlling the false discovery rate of windows that do have regulatory activity (positive betas). In the permuted data, by contrast and because of the randomization we impose, test statistics are centered around 0 and essentially symmetrical (approximately equally likely to be positive or negative). Retaining all p-values to construct the null therefore leads to highly miscalibrated false discovery rates because the distribution of observed values is skewed towards smaller values— because of windows with “significantly” no regulatory activity—compared to the permuted data. We address that problem by using only positive betas from the permutations.

i. The interpretation of the overlap of Dex-response windows with CpGs sites associated with early adversity should be revisited according to the points also mentioned in the public review and the authors may want to consider exploring additional datasets with other challenges.

Thank you, see our responses to the public review above and our revisions in lines (lines 555559). We agree that comparisons with more data sets and generation of more mSTARR-seq data in other challenge conditions would be of interest. While beyond the scope of this manuscript, we hope the resource we have developed and our methods set the stage for just such analyses.

**Reviewer #2 (Public Review):**
This work presents a remarkably extensive set of experiments, assaying the interaction between methylation and expression across most CpG positions in the genome in two cell types. To this end, the authors use mSTARR-seq, a high-throughput method, which they have previously developed, where sequences are tested for their regulatory activity in two conditions (methylated and unmethylated) using a reporter gene. The authors use these data to study two aspects of DNA methylation:1. Its effect on expression, and 2. Its interaction with the environment. Overall, they identify a small number of 600 bp windows that show regulatory potential, and a relatively large fraction of these show an effect of methylation on expression. In addition, the authors find regions exhibiting methylation-dependent responses to two environmental stimuli (interferon alpha and glucocorticoid dexamethasone).The questions the authors address represent some of the most central in functional genomics, and the method utilized is currently the best method to do so. The scope of this study is very impressive and I am certain that these data will become an important resource for the community. The authors are also able to report several important findings, including that pre-existing DNA methylation patterns can influence the response to subsequent environmental exposures.

Thank you for this generous summary!

The main weaknesses of the study are: 1. The large number of regions tested seems to have come at the expense of the depth of coverage per region (1 DNA read per region per replicate). I have not been convinced that the study has sufficient statistical power to detect regulatory activity, and differential regulatory activity to the extent needed. This is likely reflected in the extremely low number of regions showing significant activity.

We apologize for our lack of clarity in the previous version of the manuscript. Nonzero coverage for half the plasmid-derived DNA-seq replicates is a minimum criterion, but for the baseline dataset, the mean depth of DNA coverage per replicate for windows passing the DNA filter is quite high: 12.723 ± 41.696 s.d. overall, and 93.907 ± 10.091 s.d. in the windows we subjected to full analysis (i.e., windows that also passed the RNA read filter). We now provide these summary statistics in lines 148-149 and 169-170 and Supplementary file 5 (see also our responses to Reviewer 1 above). We also now show, using a rarefaction analysis, that our data set saturates the ability to detect regulatory windows based on DNA and RNA sequencing depth (new Figure 1-figure supplement 4; lines 179-184; 715-722).

1. Due to the position of the tested sequence at the 3' end of the construct, the mSTARR-seq approach cannot detect the effect of methylation on promoter activity, which is perhaps the most central role of methylation in gene regulation, and where the link between methylation and expression is the strongest. This limitation is evident in Fig. 1C and Figure 1-figure supplement 5C, where even active promoters have activity lower than 1. Considering these two points, I suspect that most effects of methylation on expression have been missed.

Thank you for pointing this out. We agree that we have not exhaustively detected methylationdependent activity in all promoter regions, given that not all promoter regions are active in STARR-seq. However, there is good evidence that some promoter regions can function like enhancers and thus be detected in STARR-seq-type assays (Klein et al., 2020). This important point is now noted in lines 187-189; an example promoter showing methylation-dependent regulatory activity in our dataset is shown in Figure 3E.

We also now clarify that Figure 1C shows significant enrichment of regulatory activity in windows that overlap promoter sequence (line 239). The y-axis is not a measure of activity, but rather the log-transformed odds ratio, with positive values corresponding to overrepresentation of promoter sequences in regions of mSTARR-seq regulatory activity. Active promoters are 1.640 times more likely to be detected with regulatory activity than expected by chance (p = 1.560 x 10-18), which we now report in a table that presents enrichment statistics for all ENCODE elements shown in Figure 1C for clarity (Supplementary file 4). Moreover, 74.1% of active promoters that show regulatory activity have methylation-dependent activity, also now reported in Supplementary file 4.

Overall, the combination of an extensive resource addressing key questions in functional genomics, together with the findings regarding the relationship between methylation and environmental stimuli makes this a key study in the field of DNA methylation.

Thank you again for the positive assessment!

**Reviewer #2 (Recommendations For The Authors):**
I suggest the authors conduct several tests to estimate and/or increase the power of the study:1. To estimate the potential contribution of additional sequencing depth, I suggest the authors conduct a downsampling analysis. If the results are not saturated (e.g., the number of active windows is not saturated or the number of differentially active windows is not saturated), then additional sequencing is called for.

We appreciate the suggestion. We have now performed a downsampling/rarefaction curve analysis in which we downsampled the number of DNA reads, and separately, the number of RNA reads. We show that for both DNA-seq depth and RNA-seq depth, we are within the range of sequencing depth in which additional sequencing would add minimal new analysis windows in the dataset (Figure 1-figure supplement 4; lines 179-184; 715-722).

2. Correlation between replicates should be reported and displayed in a figure because low correlations might also point to too few reads. The authors mention: "This difference likely stems from lower variance between replicates in the present study, which increases power", but I couldn't find the data.

We now report the correlations between RNA and DNA replicates within the current dataset and within the Lea et al., 2018 dataset (Figure 1-figure supplement 6). The between-replicate correlations in both our RNA libraries and DNA libraries are consistently high (r ≥ 0.89).

3. The correlation between the previous and current K562 datasets is surprisingly low. Given that these datasets were generated in the same cell type, in the same lab, and using the same protocol, I expected a higher correlation, as seen in other massively parallel reporter assays. The fact that the correlations are almost identical for a comparison of the same cell and a comparison of very different cell types is also suspicious.

Thanks for raising this point. We think it is in reference to our original Figure 1-Figure supplement 6, for which we now provide Pearson correlations in addition to R2 values (now Figure 1-Figure supplement 8). We note that this is not a correlation in raw data, but rather the correlation in estimated effect sizes from a statistical model for methylation-dependent activity. We now provide Pearson correlations for the raw data between replicates within each dataset (Figure 1-Figure supplement 6), which for the baseline dataset are all r > 0.89 for RNA replicates and r > 0.98 for DNA replicates, showing that replicate reproducibility in this study is on par with other published studies (e.g., Klein et al., 2020 report r > 0.89 for RNA replicates and r > 0.91 for DNA replicates).

We do not know of any comparable reports in other MPRAs for effect size correlations between two separately constructed libraries, so it’s unclear to us what the expectation should be. However, we note that all effect sizes are estimated with uncertainty, so it would be surprising to us to observe a very high correlation for effect sizes in two experiments, with two independently constructed libraries (i.e., with different DNA fragments), run several years apart—especially given the importance of winner’s curse effects and other phenomena that affect point estimates of effect sizes. Nevertheless, we find that regions we identify as regulatory elements in this study are 74-fold more likely to have been identified as regulatory elements in Lea et al., 2018 (p < 1 x10-300).

4. The authors cite Johnson et al. 2018 to support their finding that merely 0.073% of the human genome shows activity (1.7% of 4.3%), but:a. the percent cited is incorrect: this study found that 27,498 out of 560 million regions (0.005%) were active, and not 0.165% as the authors report.

We have modified the text to clarify the numerator and denominator used for the 0.165% estimate from Johnson et al 2018 (lines 175-176). The numerator is their union set of all basepairs showing regulatory activity in unstimulated cells, which is 5,547,090 basepairs. The denominator is the total length of the hg38 human genome, which is 3,298,912,062 basepairs.

Notably, the denominator (the total human genome) is not 560 million—while Johnson et al (2018) tested 560 million unique ~400 basepair fragments, these fragments were overlapping, such that the 560 million fragments covered the human genome 59 times (i.e., 59x coverage).

b. other studies that used massively parallel reporter assays report substantially higher percentages, suggesting that the current study is possibly underpowered. Indeed, the previous mSTARR-seq found a substantially larger percentage of regions showing regulatory activity (8%). The current study should be compared against other studies (preferably those that did not filter for putatively active sequences, or at least to the random genomic sequences used in these studies).

We appreciate this point and have double checked comparisons to Johnson et al., 2018 and Lea et al., 2018. Our numbers are not unusual relative to Johnson et al., 2018 (0.165%), which surveyed the whole genome. Also, in comparing to the data from Lea et al., 2018, when processed in an identical manner (our criteria are more stringent here), our values of the percent of the tested genome showing significant regulatory activity are also similar: 0.108% in the Lea et al., 2018 dataset versus 0.082% in the baseline dataset. Finally, our rarefaction analyses (see our responses above) indicate that we are not underpowered based on sequencing depth for RNA or DNA samples. We also note that there are several differences in our analysis pipeline from other studies: we use more technical replicates than is typical (compare to 2-5 replicates in Arnold et al., 2013; Johnson et al., 2018; Muerdter et al., 2018), we measure DNA library composition based on DNA extracted from each replicate post-transfection as opposed to basing it on the pre-transfection library: [Johnson et al., 2018], and we use linear mixed models to identify regulatory activity as opposed to binomial tests [Johnson et al., 2018; Arnold et al., 2013; Muerdter et al., 2018].

I find it confusing that the four sets of CpG positions used: EPIC, RRBS, NR3C1, and random control loci, add up together to 27.3M CpG positions. Do the 600 bp windows around each of these positions sufficient to result in whole-genome coverage? If so, a clear explanation of how this is achieved should be added.

Thanks for this comment. Although our sequencing data are enriched for reads that cover these targeted sites, the original capture to create the input library included some off target reads (as is typical of most capture experiments, which are rarely 100% efficient). We then sequenced at such high depth that we ultimately obtained sequencing coverage that encompassed nearly the whole genome. We now clarify in the main text that our protocol assesses 27.3 million CpG sites by assessing 600 bp windows encompassing 93.5% of all genomic CpG sites (line 89), which includes off-target sites (line 149).

A scatter plot showing the RNA to DNA ratios of the methylated (x-axis) vs unmethylated (y-axis) library would be informative. I expect to see a shift up from the x=y diagonal in the unmethylated values.

We have added a supplementary figure showing this information, which shows the expected shift upwards (Figure 1-figure supplement 9).

Another important figure missing is a histogram showing the ratios between the unmethylated and methylated libraries for all active windows, with the significantly differentially active windows marked.

We have added a supplementary figure showing this information (Figure 1-Supplementary Figure 10).

Perhaps I missed it, but what is the distribution of effect sizes (differential activity) following the various stimuli?

This information is provided in table form in Supplementary Files 3, 10, and 11, which we now reference in the Figure 2 legend (lines 365-366).

Minor changesIt is unclear what the lines connecting the two groups in Fig.3C represent, as these are two separate groups of regions.

We now clarify in the figure legend that values connected by a line are the same regions, not two different sets of regions. They show the correlation between DNA methylation and gene expression at mSTARR-seq-identified enhancers in individuals before and after IAV stimulation, separately for enhancers that are shared between conditions (left) versus those that are IFNAspecific (right). The two plots therefore do show two different sets of regions, which we have depicted to visualize the contrast in the effect of stimulation on the correlation on IFNA-specific enhancers versus shared enhancers. We have revised the figure legend to clarify these points (line 458-460).

L235-242 are unclear. Specifically - isn't the same filter mentioned in L241-242 applied to all regions?

Yes, the same filter for minimal RNA transcription was applied to all regions. We have modified the text (lines 264-265, 271, 275-277) to clarify that the enrichment analyses were performed twice, to test whether the target types were: (1) enriched in the dataset passing the RNA filter (i.e., the dataset showing plasmid-derived RNA reads in at least half the sham or methylated replicates; n = 216,091 windows) and (2) enriched in the set of windows showing significant regulatory activity (at FDR < 1%; n = 3,721 windows).

To improve cohesiveness, the section about most CpG sites associated with early life adversity not showing regulatory activity in K562s can be moved to the supplementary in my opinion.

Thank you for this suggestion. Because ELA and the biological embedding hypothesis (via DNA methylation) were major motivations for our analysis (see Introduction lines 42-48; 75-79), and we also discuss these results in the Discussion (lines 518-520), we have respectfully elected to retain this section in the main manuscript. We have added text in the Discussion explaining why we think experimental tests of methylation effects on regulation are relevant to the literature on early life adversity (lines 520-522), and have added discussion on limits to these analyses (lines 527-533).

References:

Arnold CD, Gerlach D, Stelzer C, Boryń ŁM, Rath M, Stark A (2013) Genome-wide quantitative enhancer activity maps identified by STARR-seq. Science, 339, 1074-1077.

Cecil CA, Zhang Y, Nolte T (2020) Childhood maltreatment and DNA methylation: A systematic review. Neuroscience & Biobehavioral Reviews, 112, 392-409.

Dubois M, Louvel S, Le Goff A, Guaspare C, Allard P (2019) Epigenetics in the public sphere: interdisciplinary perspectives. Environmental Epigenetics, 5, dvz019.

Eisenberger NI, Cole SW (2012) Social neuroscience and health: neurophysiological mechanisms linking social ties with physical health. Nature neuroscience, 15, 669-674.

Houtepen L, Hardy R, Maddock J, Kuh D, Anderson E, Relton C, Suderman M, Howe L (2018) Childhood adversity and DNA methylation in two population-based cohorts. Translational Psychiatry, 8, 1-12.

Johnson GD, Barrera A, McDowell IC, D’Ippolito AM, Majoros WH, Vockley CM, Wang X, Allen AS, Reddy TE (2018) Human genome-wide measurement of drug-responsive regulatory activity. Nature communications, 9, 1-9.

Klein JC, Agarwal V, Inoue F, Keith A, Martin B, Kircher M, Ahituv N, Shendure J (2020) A systematic evaluation of the design and context dependencies of massively parallel reporter assays. Nature Methods, 17, 1083-1091.

Koss KJ, Gunnar MR (2018) Annual research review: Early adversity, the hypothalamic–pituitary– adrenocortical axis, and child psychopathology. Journal of Child Psychology and Psychiatry, 59, 327-346.

Marzi SJ, Sugden K, Arseneault L, Belsky DW, Burrage J, Corcoran DL, Danese A, Fisher HL, Hannon E, Moffitt TE (2018) Analysis of DNA methylation in young people: limited evidence for an association between victimization stress and epigenetic variation in blood. American journal of psychiatry, 175, 517-529.

Muerdter F, Boryń ŁM, Woodfin AR, Neumayr C, Rath M, Zabidi MA, Pagani M, Haberle V, Kazmar T, Catarino RR (2018) Resolving systematic errors in widely used enhancer activity assays in human cells. Nature methods, 15, 141-149.